# What Tweets and YouTube comments have in common? Sentiment and graph analysis on data related to US elections 2020

**Alexander Shevtsov**[1,2], **Maria Oikonomidou**[1,2], **Despoina Antonakaki**[1]*,
**Polyvios Pratikakis**[1,2], **Sotiris Ioannidis**[1,3]

1 Institute of Computer Science, Foundation for Research and Technology, Vassilika Vouton, Heraklion, Crete, Greece, 2 Computer Science Department - University of Crete, Voutes Campus, Heraklion, Crete, Greece, 3 School of Electrical and Computer Engineering, Technical University of Crete, University Campus, Akrotiri, Chania, Greece

* despoina@ics.forth.gr

**Data Availability Statement:** In order to obtain the dataset used for the analysis described in this study, we follow the Twitter API restrictions and do

## Abstract

Most studies analyzing political traffic on Social Networks focus on a single platform, while campaigns and reactions to political events produce interactions across different social media. Ignoring such cross-platform traffic may lead to analytical errors, missing important interactions across social media that e.g. explain the cause of trending or viral discussions. This work links Twitter and YouTube social networks using cross-postings of video URLs on Twitter to discover the main tendencies and preferences of the electorate, distinguish users and communities' favouritism towards an ideology or candidate, study the sentiment towards candidates and political events, and measure political homophily. This study shows that Twitter communities correlate with YouTube comment communities: that is, Twitter users belonging to the same community in the Retweet graph tend to post YouTube video links with comments from YouTube users belonging to the same community in the YouTube Comment graph. Specifically, we identify Twitter and YouTube communities, we measure their similarity and differences and show the interactions and the correlation between the largest communities on YouTube and Twitter. To achieve that, we have gather a dataset of approximately 20M tweets and the comments of 29K YouTube videos; we present the volume, the sentiment, and the communities formed in YouTube and Twitter graphs, and publish a representative sample of the dataset, as allowed by the corresponding Twitter policy restrictions.

## 1 Introduction

Previous studies analyzing political content on Social Networks, mainly focus on a single platform, however, online political campaigns and interactions between users seem to take place across diverse online social networks. Online discourse is scattered across several social networks, sometimes among the same users and communities. The parallel study of the content of these social networks can reveal the political campaigns [1], comments, opinions,

not violate any terms from Twitter Developer Agreement and Policy. According to Twitter Policy, we are not allowed to share the entire dataset, but only 100K user IDs. This dataset is available here: https://zenodo.org/record/4618233#. YGGJU2Qzada. The access is open and no approval is required. We provide the directed retweet graph from the Twitter network, all user IDs from the provided retweet graph (89.479 users), all video IDs (vid) extracted from the election related tweets (39.203 video ids) and the directed comment graph.

**Funding:** This document is the results of the research project co-funded by the European Commission, project CONCORDIA, with grant number 830927 (EUROPEAN COMMISSION Directorate-General Communications Networks, Content and Technology) and by the European Union and Greek national funds through the Operational Program Competitiveness, Entrepreneurship and Innovation, under the call RESEARCH - CREATE - INNOVATE (project ode: T1EDK-02857 and T1EDK-01800). The funders had no role in study design, data collection and analysis, decision to publish, or preparation of the manuscript.

**Competing interests:** The authors have declared that no competing interests exist.

tendencies, and beliefs of the electorate [2, 3]. For instance, social media played a very important role during the 2016 US elections [1, 4–6]. However, only a few studies focus on more than one social networks at a time [7, 8], thus possibly missing important interactions.

Users on Twitter and YouTube form tight communities. Twitter homophily has been noticed in communities [9], in hashtags [10], on Twitter lists [11], and modelled in the follow and mention graphs [12, 13]. The analysis of this phenomenon has been studied under the prism of political context as well, like in [14–19], or support homophily between users and social media [20]. Homophily and forming of common communities on social media have been noticed on YouTube as well, as seen in several background studies [21–23].

Social media users with political interests share and seek information about politics on Twitter and there is a high probability these particular users will also search for additional information on other social networks with similar topics. This could indicate potential connections across different social platforms, especially in the case of political discourse. Consequently, the analysis of a single social network cannot reveal the entire picture of the connections across several social networks.

As supported above, online social networks allow users to form groups and communities, where the members are related to the same topic or interest. Based on that, we assume that communities are generated based on the users' preferences. Each social network has its own communities, where the users are already connected to each other, by some hidden layers of each social network. In our study, we provide an analysis of Twitter and YouTube where these layers are consisted of the features like retweets, replies, mentions - in the case of Twitter- and comments and likes -in the case of YouTube-. We already have information about the community structures of each social network, but the correlation of those communities across both of them is still not covered. From our perspective, it is very important to identify if indeed communities with similar interests are connected across different social networks.

In this study, we manage to reveal the connections between the communities across YouTube and Twitter. Through community detection in the two social networks, we reveal the interactions between the comment graph and retweet graph, respectively. To achieve this, initially, we obtain the most popular hashtags related to the US elections and gather a dataset of 19.8M tweets, for a period of four months starting from July 19 of 2020. We extract 28.343 unique YouTube video links contained in the Twitter dataset, as well as their metadata (comments, authors, etc.). We perform volume analysis, identify the main entities in the corpus and perform per-entity sentiment analysis. We study the evolving retweet graph in six sequential time periods from July to November 2020 and show the diffusion of content regarding the two candidates. The results of sentiment analysis give a higher positive sentiment towards Donald Trump on Twitter (35.2%) and YouTube (18%), in comparison to the positive sentiment expressed towards Joe Biden on Twitter (28%) and YouTube (12%). We perform community detection in the retweet graph and YouTube comment graph with the use of Louvain method [24]. With PageRank, we identify the most engaging nodes in both graphs and label the communities that contain these nodes after their name. Finally, we show the interactions between the largest, in size, communities on both social media.

This study makes the following contributions:

- Identifies the connections of the communities affiliated with D.Trump and J.Biden campaign between Twitter and YouTube.

- Reveals the interactions between the YouTube communities in the YouTube-comment graph.

- Presents the Twitter retweet communities generated based on the user retweets.

- Highlights the patterns in the comment graph that supports the linking between the Twitter Retweet graph to the YouTube comment graph.

Analyzing the sentiment is an additional step towards the processing of political content [25–29]. Sentiment analysis has been extensively applied to Twitter traffic, usually regarding a specific event. Through sentiment analysis, we can visualize the variation of the sentiment of the electorate, during a political event [30, 31], a company event [32], a product's reviews [33], or model the public mood and emotion and connect tweets' sentiment features with fluctuations with real events [34]. The main task of these works is to predict elections, classify the electorate and distinguish posts towards one political party or ideology, although it has been addressed in many works that Twitter is not suitable for election prediction [35, 36].

## 1.1 Background

**1.1.1 Homophily, communities on Twitter, YouTube.**   Existing studies towards the analysis of the interactions across different social networks during the elections period are used for diverse goals. They mainly focus on the observation and investigation of the online discourse, highlighting the main activity of the candidates and the evolution of their online presence [37], evaluate how the candidates are influenced by SNs [38], or investigate whether social networks contribute in the democratization of our political systems, studying online propaganda, or the topics of the content in YouTube videos related to campaigns [39].

There is a plethora of studies on homophily on Twitter and YouTube on political content, analyzing the communities that are formed on social media and content generated by these platforms. For example in [40] authors manage to analyse 8.9 M tweets during the 2015 Spain Elections. This study identifies a specific category of users, the so-called "party evangelists" and explores the activity effect on them regarding the general political conversation. In [41] they analyze the communications of alt-right supporters during the 2018 US-midterm elections. After they obtain a dataset of 52903 tweets posted by 123 alt-right Twitter accounts (from 30/10 until 13/11 2018), they apply community detection (Clauset-Newman-Moore greedy modularity maximization method) in the retweet graph, topic extraction of the communities and demonstrate the communities categorized by the topic discussed among them. In [42] they study homophily in journalists' interactions by analyzing a dataset of 600,000 tweets sent by 2908 Australian journalists in a period of one year. They conclude that journalists mainly interact with other journalists of the same gender, working environment, and location forming a bubble.

In [43] they investigate the structure of the "Occupy Wall Street" movement on Twitter and YouTube through a dataset of 328 identified users including their posts, the number of followers, followees tweets, and favorites of users, as well as the corresponding YouTube videos with the keyword "Occupy Wall Street". The results show that both social networks were coordinated by US users and specifically, a loosely connected network coordinated around central hub users on Twitter while on YouTube the network was organized by anonymous users. In [44] they analyse a dataset of 238,967 tweets around the #gamergate, associated with an episode "Law & Order: Special Victims Unit" that initiated an online activity and controversy and explore the challenges of accounting on the cultural dynamics of Twitter and YouTube, focusing on the key media objects like images videos and tags.

Community detection has been used towards the discovery of relations between different and heterogeneous social networks like in [7], where they build a heterogeneous network with undirected and directed edges, to simulate a social network and study overlapping community detection. Also, in [45] they apply a method of converting multi-relational networks to single-

relational networks on synthetic and real-world data (DBLP). Background work in community detection across Twitter with other real social network is limited.

**1.1.2 Political content analysis on Twitter.**   The background work specifically during election periods on Twitter includes volume analysis of the number of tweets and comparison with the results of the elections, while they apply topic analysis and show the co-occurrence between terms or candidates within the tweets [3, 46–48]. Other works focus on the classification of social media users to separate Democrats and Republicans by using machine learning techniques on features derived from user information [49], or by automatically constructing user profiles in a dataset of scraped lists of users in the Twitter directories WeFollow and Twellow [50]. For example in [51] in a categorized dataset of four samples covering the period of January 2013 until December 2014, they apply unsupervised learning to classify the tweets topics posted by the legislators and citizens and reveal that the former follows a discussion of public issues and are potentially more interacting with their supporters that to the general public. In [52] on a dataset of 13 million followers of @realDonaldTrump on 8 November 2016, they apply a clustering analysis and show the following patterns of the electorate.

Studies that focus on network analysis on Twitter, during periods of election study also homophily [13, 14, 53, 54], apply stochastic link structure analysis and show the party interactions [55], apply community detection [56, 57], analyze the networks of mentions and retweets [57–60] and estimate users' partisan preference, while in [60] they study the activity, emotional content, and interactions of political parties and politicians. Finally, in [61] they study the structure of the conversations on Twitter between political, media, and citizen agents, while the analysis shows a decentralized and loosely knit network, during elections in Belgium.

Studies are focusing on the prediction of the election outcome([62, 63]), like in [48] where there are also studying whether Twitter can be used for political deliberation and mirrors offline political sentiment, during the German federal election.

There is a plethora of studies that have used sentiment analysis in the political domain [64, 65], either for group polarization [13, 66], study specific events or personalities like the Arab spring [67] or Hugo Chavez [68]. For example, in [30] they study a dataset of 36 million tweets on the 2012 U.S. presidential candidates and apply a real-time analysis of public sentiment. Using the Amazon Mechanical Turk, they label the dataset with the tweets' sentiment (positive, negative, neutral, or unsure), in order to apply statistical classification (Naive Bayes model on uni-gram features) on a training set consisting of nearly 17.000 tweets (16% positive, 56% negative, 18% neutral, 10% unsure). Also, in [69] they attempt to understand the broader picture of how Twitter is used by party candidates, understand the content and the level of interaction by followers. More detailed background on Twitter Sentiment Analysis methods can be found on surveys like [70–73].

Sentiment Analysis on Twitter is not limited to one language; there are works studying multiple language datasets [74–80]. Recent studies in sentiment analysis using Deep Convolutional Neural Networks [74, 81–86]. Our work focuses on a single language (English) which is the dataset based on. Finally, some studies incorporate the use of Twitter features, like emoticons [87–90].

Sentiment in these analyses is represented by a variable with values like 'Positive', 'Negative' and 'Neutral', or even more specific ('Happy','Angry', etc.). Each word in the corpus can be assigned with more than one sentiment ('Positive' and 'Negative'). Other metrics that can be measured in this analysis, are 'subjectivity' and 'polarity', where the first one is defined as the ratio of 'positive' and 'negative' tweets to 'neutral' tweets, while the second is defined as the ratio of 'Positive' to 'Negative' tweets.

A necessary step of sentiment analysis is 'text normalization', an initial preprocessing of the corpus to extract the lexical features that can significantly affect the performance [91–93]. The

steps of the preprocessing include tokenization, expansion of abbreviations, and removal of stop words (URLs, mentions, etc.).

A very common step that is also used is to incorporate a lexicon, specially made for the domain of the dataset [3, 94]. In [95] they obtain three different corpora of tweets and explore the usage of linguistic features towards sentiment analysis. In [96] they are adopting a lexicon-based method on diverse Twitter datasets. In [97] they conduct a study of sentiment analysis on a dataset of 26,175 general Bulgarian tweets. Through feature selection and classification (binary SVM), they show that the negative sentiment predominated before and after the election period. In the current study, we are not compiling a specially made lexicon, because we do not have a language barrier or analyzing a plethora of entities; we are just focusing on the two main candidates.

**1.1.3 Content analysis on YouTube.** YouTube analysis has been used to apply sentiment analysis in the recent US Elections 2020, on limited dataset of approximately 200 comments from YouTube [98], to discover irrelevant and misleading metadata [99], to identify spam campaigns [100], to discover extremists videos and hidden communities [101], to propagate preference information of personalised video [102], to estimate causality between user profiles [103], to spread political advertisement [39, 104, 105], and to apply opinion mining [106]. For example in [104], they explore the Senate Campaign 2008. Among other, they claim that not many people are exposed to television ads, therefore YouTube represents an increase in accessibility of the campaign messages, which favored a group to control the messages on the network. Background work on sentiment analysis on YouTube [107, 108], studies the sentiment on user comments [109, 110], identifies the trends and demonstrates the influence of real events of user sentiments [111], implements model utilizing audio, visual and textual modalities as sources of information [112] and studies the popularity indicators and audience sentiments of videos [113].

Previous studies on both Twitter and YouTube, during the elections period, investigate online propaganda strategies like in [8], or evaluate the influence of candidates via social media [38].

In our study, we focus on the discovery of the relation between YouTube and Twitter via community detection and on the link of Twitter Retweet graph with YouTube comment graph, by using tweets of posted videos, the measurement of their similarity and differences, and the interactions and the correlations between the largest communities on YouTube and Twitter. As seen above, previous work on the analysis of social media corpus during election periods like [98], does not provide this large dataset covering almost three months before the elections of political conversation (in two social media), nor does this extended analysis covering sentiment, volume and graph analysis, as well as the comparison and correlation between two social networks.

## 2 Dataset

We acknowledge the fact that Twitter and YouTube users do not fully represent the electorate during an election period, which consequently introduces a bias in the electorate dataset. Previous studies have shown that Twitter users belong to a specific age [36], social and ideology demographic group [114]. This means that public opinion is not fully expressed through social media. Regarding YouTube, the usage penetration in the United States 2020, by age group is in better shape than Twitter, since the popularity difference between the age groups differs slightly, although it attracts a younger audience, regardless of the minimum age for using the service is 13 years, in most countries [115], not necessarily part of the audience of our dataset.

Taking into consideration the bias of the dataset, the task of election prediction is challenging. However, the analysis of the specific corpus can shed light on the relationship between Twitter and YouTube. As shown in section 4.4, the task of community detection on both social graphs can contribute to the association between the communities on the YouTube comment graph with the communities on the Twitter retweet graph and the measurement of their similarities and differences.

## 2.1 Twitter

In this study, we search for all the prevalent hashtags regarding the US elections on the 3rd of November 2020 and obtain the Twitter corpus through streaming Twitter API. The acquisition of the dataset started on 19 July 2020 and finished on 3 November 2020.

This resulted in a dataset of approximately 19.8M tweets, containing 4.5M users, with the most prevalent hashtags being #Trump2020, #vote and #Election2020. In Table 1 we can see the most popular hashtags, sorted by the number of tweets within which they are contained and in S1 Appendix, on Table 5 in S1 Appendix the whole list of hashtags used in our analysis. In Fig 1 we show the number of daily tweets and comments on YouTube, where we notice a peak on September 29, potentially explained by the first debate [116] and of course the second higher peak for Twitter on the day of the elections.

We considered that 19 July 2020 date was a reasonable starting point for collecting our dataset since the number of tweets in the corresponding hashtags we collect begin to accumulate a significant number, as well as the semantics of the content, started to be more relevant to the conversation related to the elections. Regarding the completeness of the content covered by our dataset, we acknowledge the fact that additional minor hashtags may exist during that period, that were not crawled and included in our corpus. We consider our dataset the complete online discourse, since we got the majority of the hashtags available in that period before the elections. Additionally, there was an overlap between the hashtags in the election discourse and these hashtags were cross-referenced in the tweets. For example, some tweets included the popular hashtags (#Vote) and the minor hashtags were also mentioned in the same tweet. This tweet is also included in our dataset because of #Vote. Additionally, we included only the general hashtags and not the ones that were in favor of a particular candidate. We try to include only two hashtags that are in favor of each candidate, in order to keep a balance between them and not introduce a bias towards one of them. Finally, the main amount of the political conversation was gathered in the popular hashtags and the rest of them do not contain a significant amount of tweets. In the S1 Appendix, in Table 5 of S1 Appendix we show the list of 20

**Table 1. The 10 most popular hashtags in our dataset.**

| Hashtag | Tweets count |
|---|---|
| #vote | 7.196.981 |
| #trump2020 | 3.913.969 |
| #election2020 | 3.535.323 |
| #biden | 1.569.981 |
| #trump | 959.048 |
| #bidenharris2020 | 866.120 |
| #debate2020 | 855.543 |
| #votebluetosaveamerica | 837.089 |
| #maga | 697.381 |
| #trumphascovid | 581.456 |

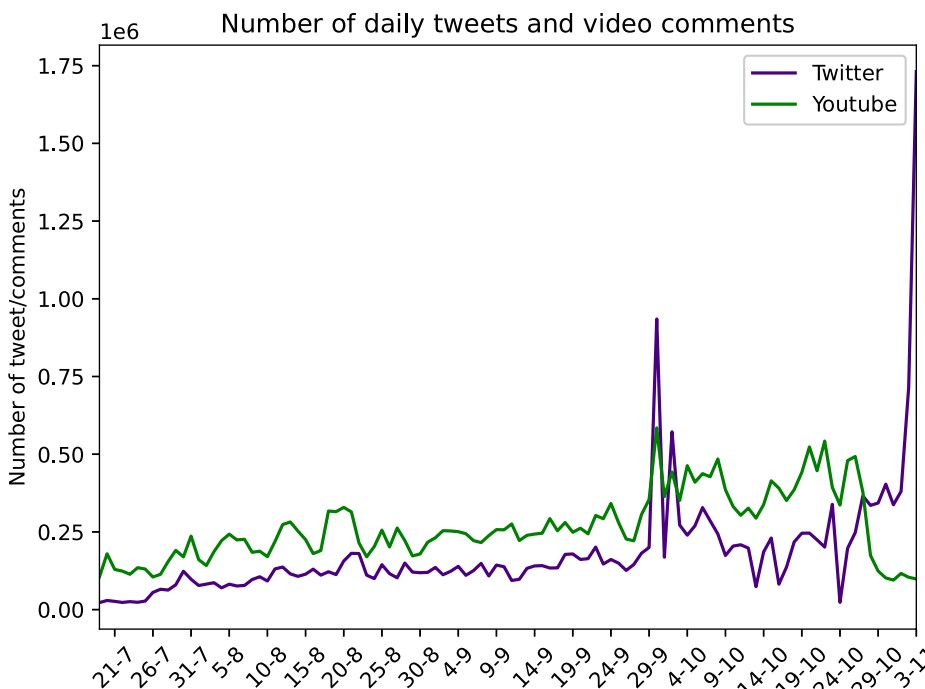

**Fig 1. Number of daily tweets and comments on YouTube videos.**

most popular hashtags in our dataset. The entire list contains 585.486 unique entries of hashtags, which is too long to be added to the manuscript.

Fig 1 shows the total number of tweets per day, for every hashtag contained in our dataset.

## 2.2 YouTube

From the election tweets, we extracted all the contained YouTube video links. We followed these links and ended up with 39.203 videos. Through the YouTube Data API, we obtained all the publicly available data regarding these videos. The accessible data used in this study for each video are:

1. Category where it belongs (e.g. News & Politics, Entertainment, Music, etc.),

2. Text and author of each selected comment and,

3. YouTube channel that posted the video.

In Fig 2 we see how many videos belong to each category. In this study, we focus on the election's topic, so we filtered out the videos that do not belong to the following categories: News & Politics, People & Blogs, and Entertainment. The filtering led to a dataset of 28.343 videos.

From the 28.343 videos, we gathered all the comments and their replies generated between 19/7/20 and 3/11/20. This resulted in a dataset of 8.476.193 unique commenters and 39.266.355 comments and replies. Fig 1 shows the total number of comments and replies per day, related to the elections. We notice an increase in the number of comments from July to November, with diurnal patterns, a peak on September 29 on the first debate [116] and of course the second peak for Twitter on the day of the elections.

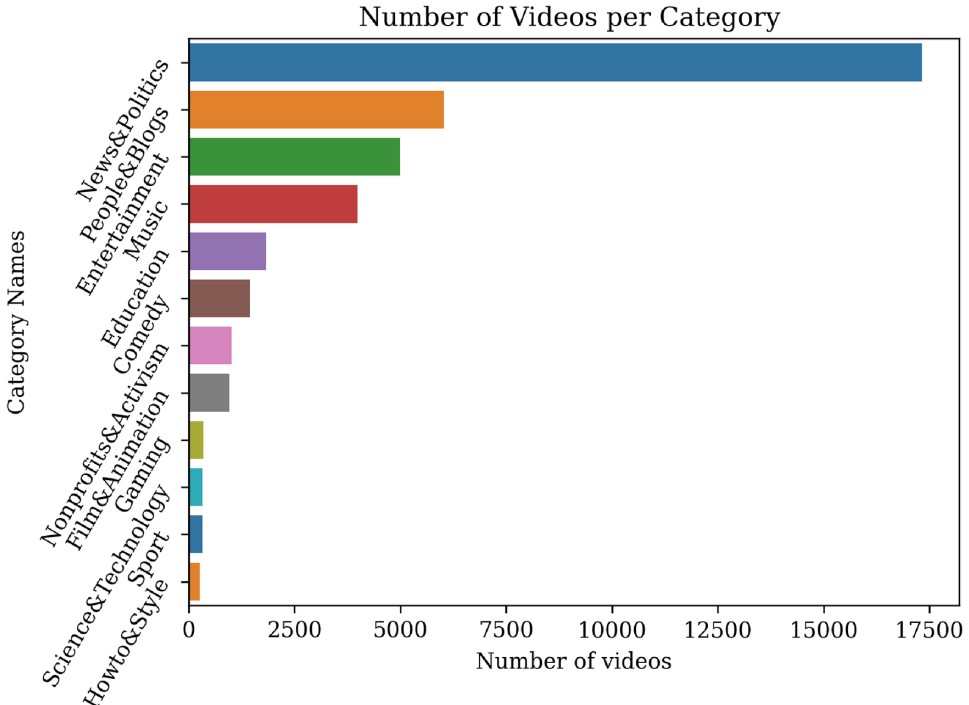

**Fig 2. Number of videos per YouTube category.**

## 2.3 Data availability

To obtain the dataset used for the analysis described in this study, we follow the Twitter and YouTube API restrictions and do not violate any terms from the Developer Agreement and Policies. According to Twitter Policy, we are not allowed to share the entire dataset, but only 100K user IDs. This dataset is available here: zenodo link. We provide the directed retweet graph from the Twitter network, all user IDs from the provided retweet graph (89.479 users), all video IDs (vid) extracted from the election-related tweets (39.203 video ids), and the directed comment graph.

The public dataset is composed of the collected data, split into two separate parts, the Twitter and YouTube parts. Each part contains limited information, in order to protect the privacy of users and respect the limitations of the social platform policies.

**2.3.1 Twitter.** The Twitter part, contains the user_IDs from the retweet graph of the users contained in our dataset, with respect to Twitter policy and limits. This retweet graph can be utilized, in order to identify the retweet relationship between the users during the US Election 2020. In this graph, nodes represent the users in the graph, while edges represent the retweet relationship between two users, accompanied by an edge weight representing the number of retweets between the particular pair of users. Also, the described nodes in the retweet graph are used, in order to collect user information, such as the user posts and user profiles. For these purposes, we used a crawler that respects the Twitter API policy, which is available here: tWawler. This crawler allows the retrieval of the user posts from the user timeline and the profile information based on the provided User_IDs, from Twitter API. In order to keep only the posts that are correlated with the US Election topic, the collection should be filtered based on the hashtags listed in Table 5 in S1 Appendix.

According to Twitter policy, we are not permitted to store and distribute user objects and information that have been removed or suspended from Twitter. Consequently, the shared dataset contains only the tweets that contain the specified hashtags. We do not provide the user information (user objects) and the timeline posts for the accounts, since this information is removed or can be removed in the future by Twitter. Thus, the shared dataset includes only the User_IDs which allows the collection of the publicly available information through the Twitter API, without violating the Twitter policy. For each User_ID we also provide a You-Tube video ID that was posted by the user during the elections period. Based on those video IDs, it is possible to collect the users who post the comments and replies to each video with the usage of YouTube API.

**2.3.2 YouTube.** The YouTube dataset contains a list of video IDs, extracted from the related tweets, described above. For this part, we provide the graph that represents the relationship between two YouTube users commenting on a specific channel. To gather the data needed for this collection part and follow YouTube API's policy restrictions, we used the scripts available at YouTube Data API. Each video on YouTube is categorized based on its content. From the videos obtained from the tweets, we exclude the ones that did not belong to the categories News & Politics, People & Blogs, and Entertainment. For the resulting filtered videos, we gathered all their comments and authors from 19 July 2020 to 3 November 2020, along with the channel that owns them. Initially, to construct the comment graph for each channel, we gather for each filtered video_ID all the user User_IDs from YouTube that have commented on it. Next, we convert the video_IDs to channel_IDs pointing to the channel that owns the video. The result is the graph containing the users and the comments on each channel.

# 3 Methodology

## 3.1 Text preprocessing

In order to apply sentiment analysis, we follow a prerequisite set of steps for preprocessing the corpus (tweets - YouTube comments); by removing punctuation symbols, text emojis and URLs; and by modifying mentions and hashtags (remove starting characters of '@' and '#'). This procedure removes the text noise and finally allows the identification of the entity that was discussed by the users. The next step includes the transformation of the text to lower case and tokenization of the collected tweets. Then, we perform the lemmatization of each token. This technique normalizes the inflected word forms. As a result of the previous steps, this sentiment analysis will be performed on lower-cased and normalized sentences.

## 3.2 Sentiment analysis

There is a plethora of previous studies as mentioned in section 1.1 that analyses the sentiment on Twitter. Vader is broadly used in this domain, as shown in [117–121].

In our implementation of sentiment analysis, we utilize Vader [122] sentiment analysis model from python NLTK library [123]. We choose this particular implementation because it is especially attuned to the sentiment expressed on social media. Vader uses a list of lexical features that are labeled according to their semantic orientation. Thanks to the implementation simplicity, sentiment analysis execution time remains low. Utilizing the already developed sentiment solution, we need to perform text filtering and parse our data to the SentimentIntensityAnalyzer function that returns the scores of positive, negative, and neutral sentiment types. In our analysis, we compute such sentiment for each collected tweet and comment in our database and summarize those sentiment scores daily for each particular user. We compute two values of daily sentiment scores per user: summary, where the daily user sentiment is a sum

product of scores of tweets that was created by the user each day; and the average score where the summary score is divided by the number of tweets where a particular entity was mentioned by the user.

The sentiment analysis of the corpus develops only the emotion vector of a particular sentence and not the emotion of a particular entity. In order to continue our pipeline, we will need the sentiment of each entity. In the next subsection, we explore how we identify the entities.

### 3.3 The entities and their sentiment

In order to identify the entity that was contained in each tweet, we generate the set of keywords, as shown in Table 2, for the particular dataset entities ('Trump' and 'Biden'). The keywords are being searched in lower case in order to avoid any misspellings and use upper-lower case writing style. We are matching those keywords in each tweet from the Twitter dataset and in each comment in the YouTube dataset, to identify whether the entity was used on the specific tweet/comment text. The entities of J. Biden and D. Trump were identified with a computed set of keywords (including the hashtags enriched with the candidates' names and the parties they are representing). For example hashtag #VoteBlue, as a text does not contain the word Biden but is associated with J. Biden campaign. We manage to recognize it because of our keywords shown in the Table 2. For this reason, we use as keywords the hashtags correlated with each candidate. This approach increases the accuracy of the association of a particular tweet/user with the described entities.

Each time the keyword is found in a particular tweet or comment, we assign the sentiment values on those entities to the user who posted this particular tweet. These entity sentiments are assigned to the users daily, since we are interested in the identification of the user/community dynamically and present how the entity sentiment evolves day by day.

## 4 Results

### 4.1 Volume analysis

In this Section, we include several volume measures derived from our dataset. Initially, we perform a volume analysis on the whole corpus of the tweets. In Fig 3 we plot the cumulative distribution of the daily tweets.

In Fig 1 we plot the number of daily tweets and video comments, where we notice the diurnal increases of posts and the massive increase of the number of tweets, on the day of the elections. In Fig 4 we can see the cumulative distribution of the daily tweets per hashtag, where we notice the prevailing hashtags of #Debate2020, #Trump, #BidenHarris2020, #VoteBluetoSaveAmerica and #Biden.

We plot the daily active users per entity, for both Twitter and YouTube, in Fig 5 for each entity of 'Trump' and 'Biden' and we notice that the total number of users is increasing from July to November, the top hashtag is the #Debate 2020 and that the users posting on Twitter and commenting on YouTube for Trump, exceed the users for Biden. The apogee of user activities is presented on the dates of the 29th of September and on the 2nd of October. These user

**Table 2. The complete list of all entities with the corresponding keywords that were used for each one.**

| Entity | Key words |
|---|---|
| Trump | trump, donald, donaldtrump, trump2020, votetrumpout, trumpislosing, dumptrump, nevertrump, republican |
| Biden | biden, joseph, bluewave2020, votebluetosaveamerica, voteblue, ridinwithbiden, neverbiden, democrat |

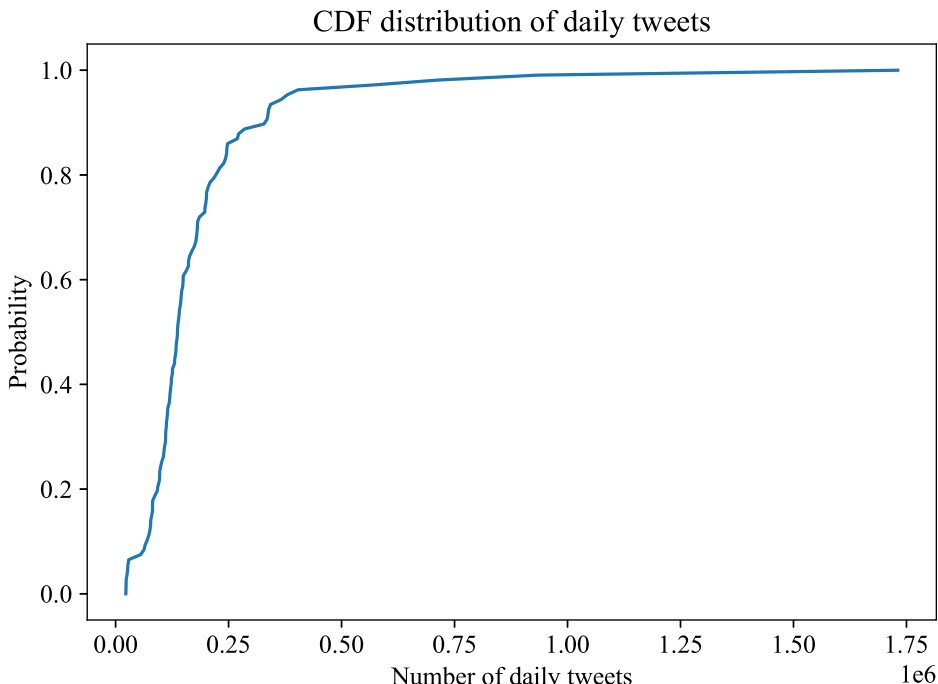

**Fig 3. CDF distribution of tweets per day.**

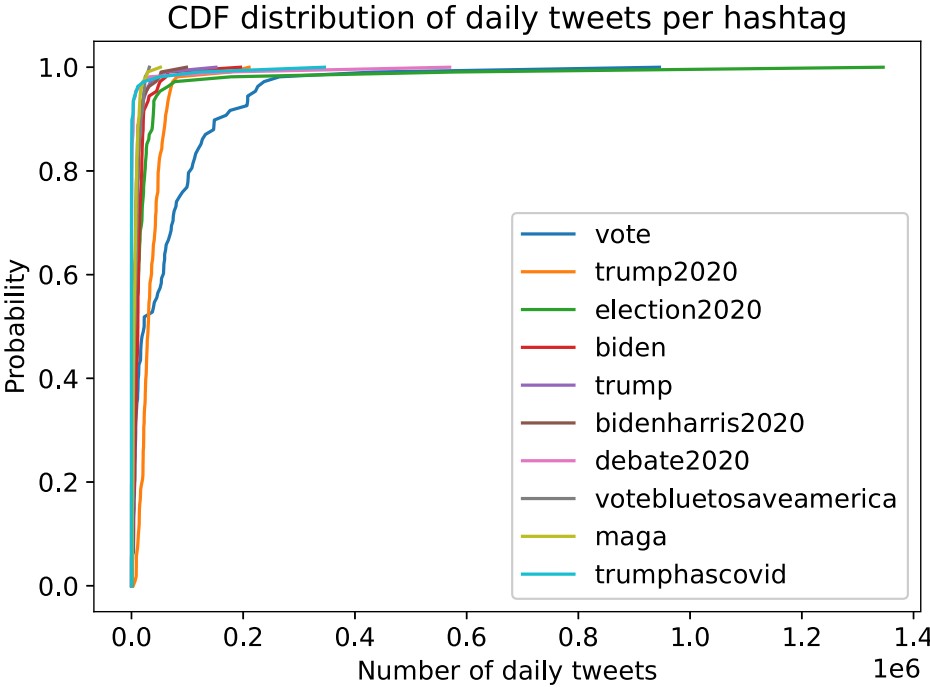

**Fig 4. CDF distribution of the daily tweets per hashtag.**

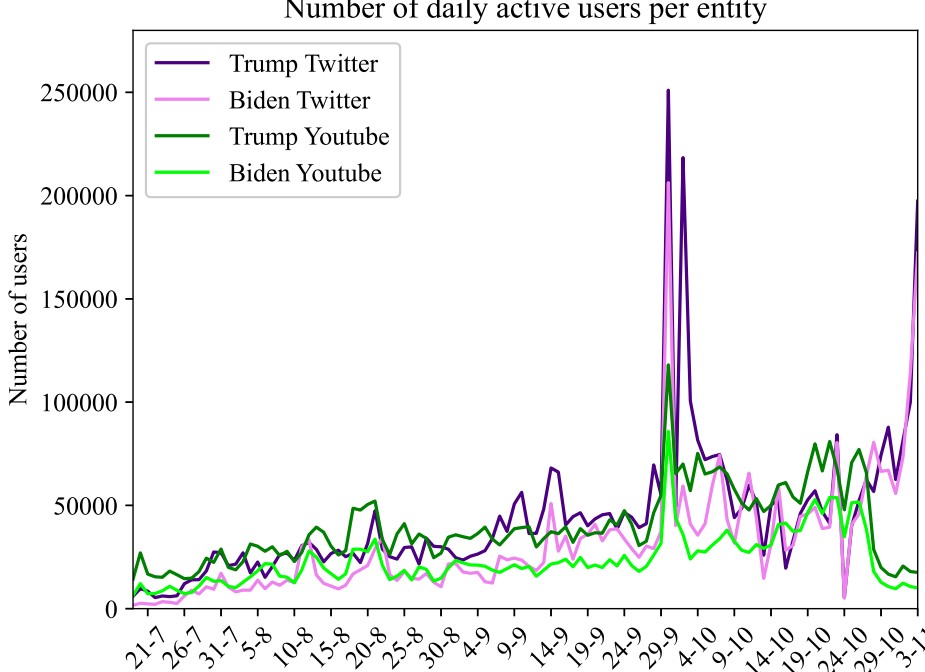

Number of daily active users per entity

**Fig 5. In this figure, we plot the number of daily users that post YouTube comments and tweets where particular entities are discussed.**

activity anomalies are explained in more detail in Section: 4.4, where the increased interest of users is correlated with election events. We also notice that the total Twitter traffic overcomes the total YouTube traffic.

## 4.2 The retweet graph

In Figs 6–11 we plot the retweet graph. This graph is developed based on the retweet relationships between all users in the collected dataset. We represent the retweet relation as a directed edge between two users (nodes). In those directed relations the destination node is the original user who posted the tweet and the source node is the user who actually retweeted that particular tweet. Such representation allows the identification of the user communities that share the same information source. To show how tight the relations are, we assign the number of seen retweets for a particular destination node, as an edge weight. We also perform filtering, by removing the non-significant edges, with weights less than 8. The filtering procedure reduces the noise volume of the non-significant relations and also reduces the number of edges that are not manageable for graph visualization. Before filtering, our retweet graph consisted of 2.874.090 nodes and 9.993.122 edges and after the filtering of the non-significant edges, we reduced the number of nodes to 56.853 and the number of edges to 86.668. For the entity visualization, we use 2 colors for our entities; in red we represent the entity of Trump, and in blue the entity of Biden.

We apply sentiment analysis for each day and we measure the number of tweets a user posts containing a particular entity. We use this counter to provide coloring of the user node by selecting the most popular entity of the user at each particular date. We also develop a plot (Fig 5) with the volume of users that allows us to compare the daily volume per entity. Users that don't use any entities in their tweets remain without a particular coloring.

Retweet graph date: 19/07/2020

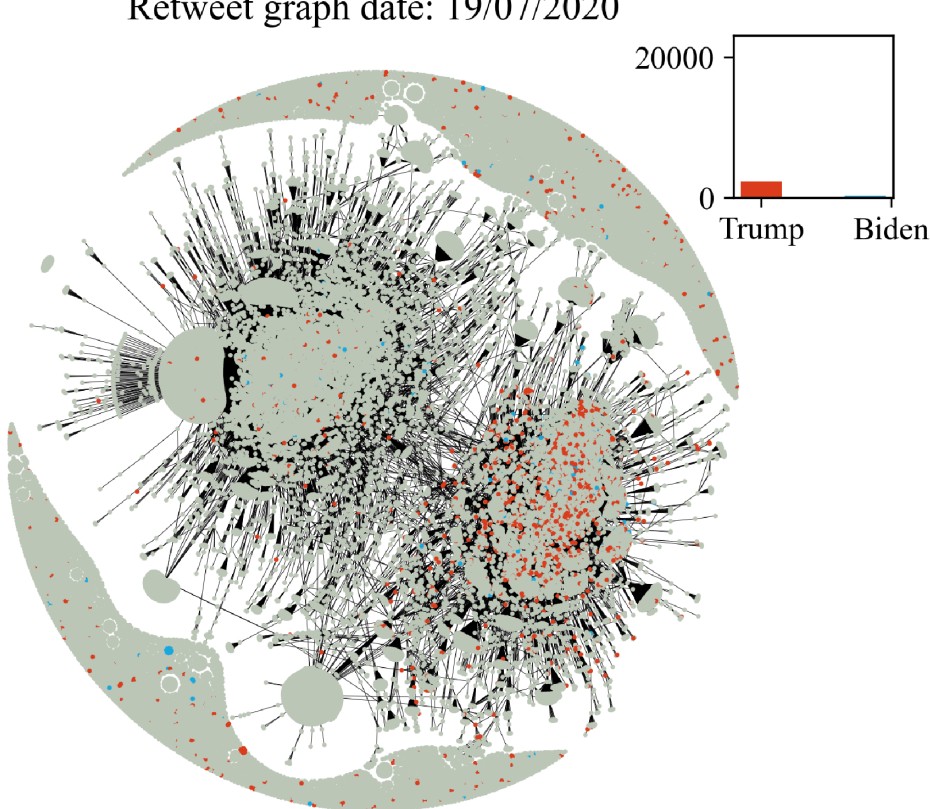

**Fig 6. Retweet graph relation between users in our dataset.** Colours represent entity activity by users. Red colour entity Trump and blue color entity Biden.

The graph plots in Figs 6–11 were generated with Gephi Furchterman Reingold layout [124], while we export Gephi generated positions and we use them to generate a daily graph with networkX python library [125]. In Figs 6–11 we notice the largest connected component acquires color in the period closer to the elections, while other cores are formed and the mentions of the two main candidates increases or in other periods decrease potentially influenced by real events like the debate (see Section 4.4). We also show how the discussion around those entities obtains more resonance in the period closer to the elections. Also, in Tables 3 and 4 we show the highest retweeted users, with the highest numbers of in-degree and out-degree respectively, with anonymized usernames.

The retweet graphs show the volume of the tweets and the relation between the nodes which represent the users. This apposition of the graphs through the whole pre-election period, demonstrates how the volume of the communication and interaction between the involved users evolves. As we expected, the density of the graph is increasing, which means the volume of the tweets increases, and the discourse is getting more intense, as we approach the period close to the elections.

The discussion around each candidate evolves through time as well. The corresponding hashtag for each candidate is mentioned as long as the electorate references them in the discourse. Of course, this does not necessarily mean a preference towards Trump or Biden. As mentioned in section 2, this analysis does not focus on the prediction of the preference of the electorate or the outcome of the elections; we expect to see how users engage in the online conversation. As we get closer to the election date, we notice some components being formed in

Retweet graph date: 01/08/2020

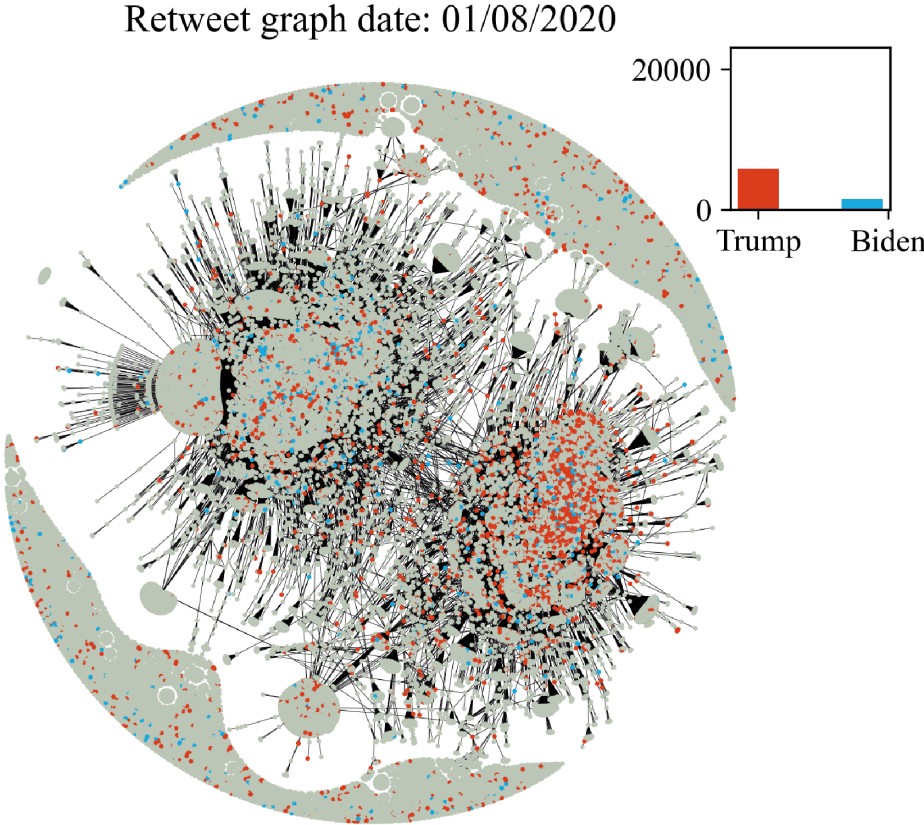

**Fig 7. Retweet graph relation between users in our dataset.** Colours represent entity activity by users. Red colour entity Trump and blue color entity Biden.

the graph that show the conversations about each candidate and the conversation about both of them in Fig 11 in the last bottom figure (Fig 11).

Additionally, we see two main graph components throughout the whole period that change the colour according to the daily events. For example, in the button left subgraph (Fig 10) we notice that the colour is red in both components, while the button right Fig 11 one is painted blue (from J. Biden). This happens because the left graph highlights the fact that Trump was infected with COVID-19 on 2/10, and the conversation about Trump was more intense and increased in terms of tweets. Also in the middle right plot we show the day after the debate, whose mixed red-blue colour makes sense if we consider the conversations this particular debate initiated. The colours as expected seem to resume back to normal on the last bottom right (3/11, which corresponds to the main two components blue and red, one for each candidate, since on this date none of the candidates seem to attract any peculiar attention.

### 4.3 Sentiment analysis

In this Section, we present the results from the sentiment analysis in our corpus, as described in 3.2. In Fig 12 we present the daily average sentiment for the entity 'Biden' and 'Trump'. The solid line is the average sentiment on YouTube comments and the dotted line is the average sentiment in the corpus of the tweets. Below zero we have the negative sentiment and above zero the positive sentiment for each social media. In Fig 13, we plot the daily overall sentiment for entity Biden and 'Trump'.

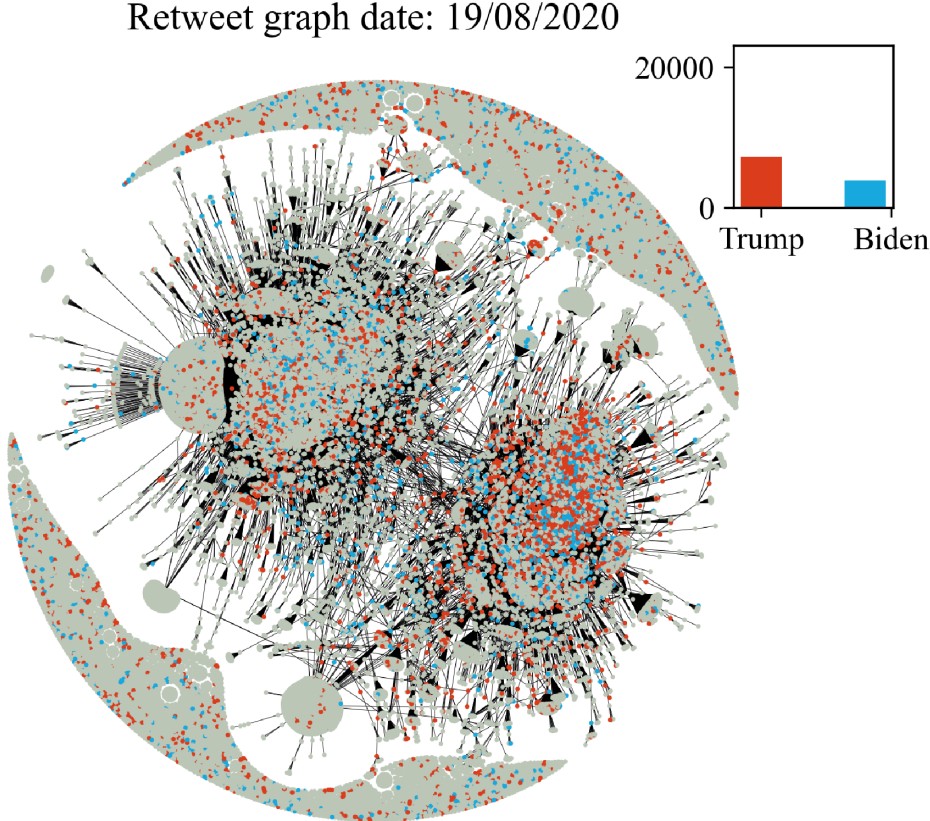

**Fig 8. Retweet graph relation between users in our dataset.** Colours represent entity activity by users. Red colour entity Trump and blue color entity Biden.

Additionally, in Fig 14 we plot the positive sentiment time series for the two sets of hashtags for each state. In order to identify the location of the user accounts, we extract this information from the corresponding field named 'location' in the Twitter user object. Additionally, we append this information with the location field in the tweet object of the Twitter API. We acknowledge the fact that the first field may be not updated by the user or the second field could be missing because the user does not share the location. This means that the location for many users could be missing. Nevertheless, this is the closest information we have from Twitter about the user location. We notice the daily fluctuations for every state per entity (blue is the entity 'Biden' and red is for 'Trump'). The juxtaposition of the time series in the form resembling an EEG makes it easier to discern localized events from nationwide Twitter traffic. The list of state abbreviations can be found here: [126].

## 4.4 Event consequences

Since our work is based on the analysis of the 2020 US Presidential elections we monitor real-world events that may trigger significant user interest on social media. Interesting examples of such events are the candidates' debate on TV (September 29 and October 22) and the date when President Donald Trump was diagnosed positively for COVID-19 (October 2). The depiction of the online conversations regarding these events is visible within our analysis. Analyzing the user engagement of these specific periods, one day before and one day after, shows whether such real-world events are connected in the virtual world of social networks.

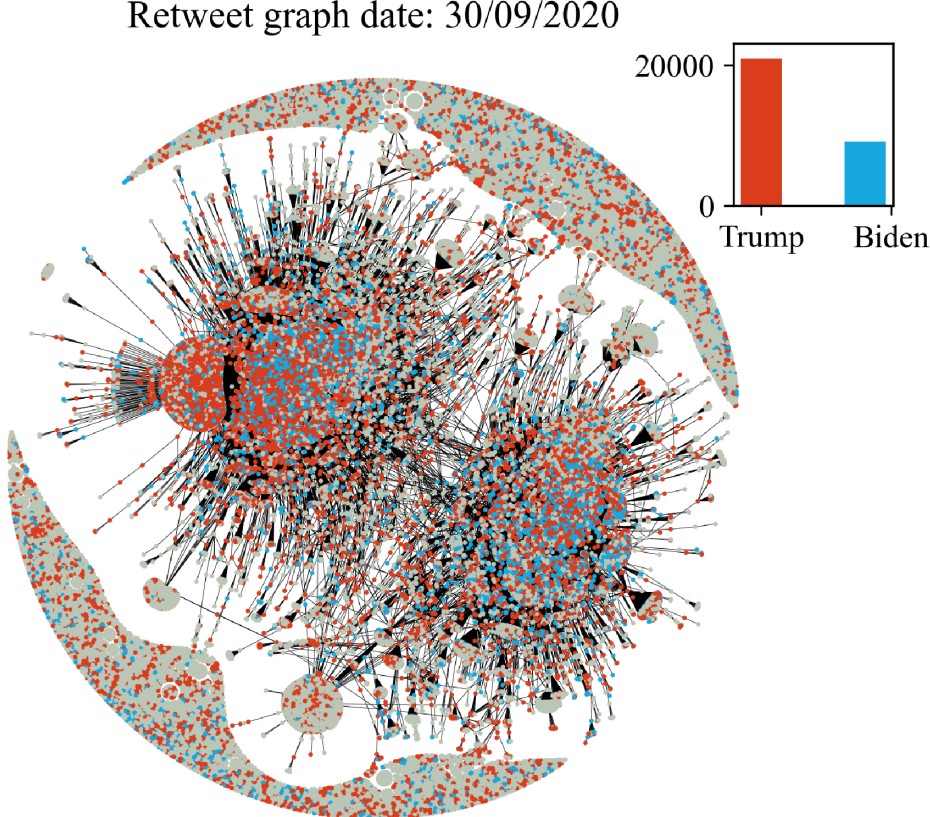

**Fig 9. Retweet graph relation between users in our dataset.** Colours represent entity activity by users. Red colour entity Trump and blue color entity Biden.

We use sentiment analysis on these specific time points of our dataset timeline to allow us to identify how the social media users react to those occasions, identify the fluctuations of the sentiment and measure the volume around each entity topic.

Our results are presented in Figs 15 and 16, where it is noticeable that the first debate and Donald Trump COVID-19 announcement events generated a high volume of user interest on social media. In the first case (debates) both entities are soared, in comparison with previous dates. At the second event, the volume of the entity 'Trump' is taking first place during the user discussions on social media by increasing the volume of tweets for the entity 'Trump' and by presenting high dissonance on sentiment values.

## 5 Linking Twitter and YouTube data

In this section, we explore the differences in the discussion and the community between the two social networks. We perform Louvain community detection on both social graphs, we associate the communities in the YouTube comment graph with the communities in the Twitter retweet graph and measure their similarity and differences. Fig 17 shows the 3-core of the YouTube comment graph, Fig 18 shows the 3-core of the Retweet graph. In both graphs, we have color-coded communities and labeled the top-Pagerank nodes of each community.

We used Gephi [124] for the analysis and visualization of Figs 17 and 18.

Fig 17 illustrates the 3-core YouTube comment graph that represents the relation of which channel has commented on which channel. Different colored areas (purple, light grey, green

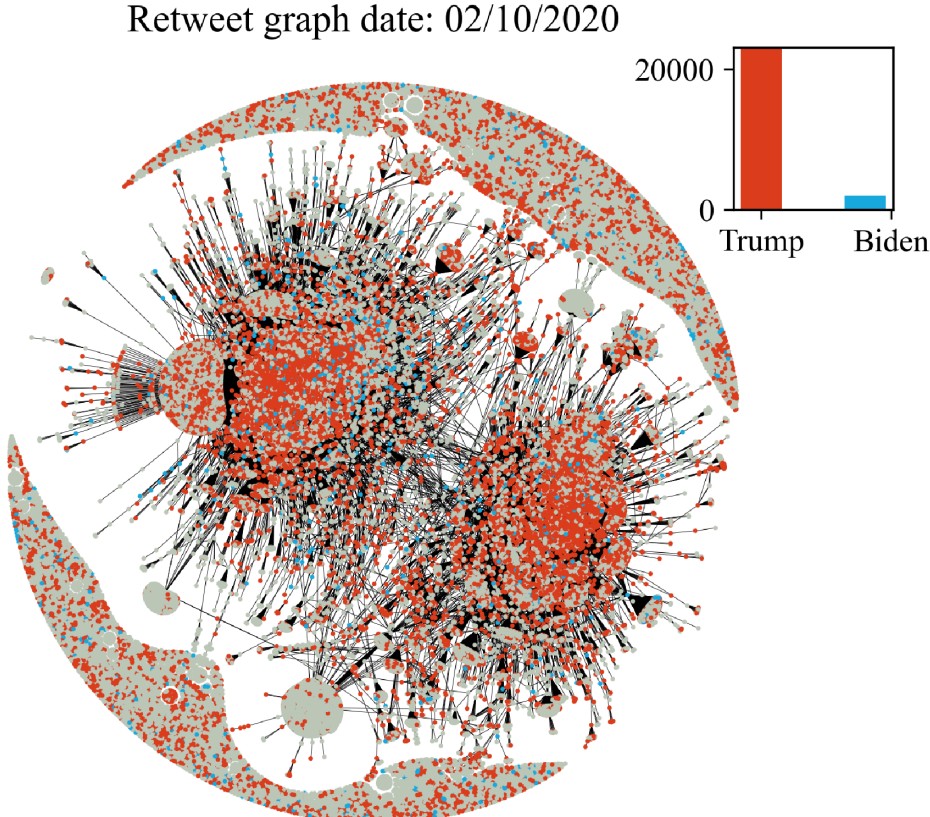

**Fig 10. Retweet graph relation between users in our dataset.** Colours represent entity activity by users. Red colour entity Trump and blue color entity Biden.

blue) in the graph represent different communities that are formed between the YouTube channels, produced by the Louvain algorithm [24]. For example purple indicates the community between the "Blaze TV", "The Officer Tatum" and the "Fox News". However "Fox News" is also between the purple and the grey, which shows the next community (formed by "Fox News" and "Donald J Trump"). These communities do not necessarily correspond to a real life community; they are a result of the Louvain algorithm of Gephi. Regarding the labels, in this figure, we also show the 13 most popular channels after running PageRank.

Similarly, in plot 18 we show the 3-core Retweet graph that represents the relation between users that have retweeted. In the graph we notice seven colored areas of different sizes, indicating different communities, that as in plot 17, do not necessarily correspond to a real life community. Also, the text labels shown, for example, "MaryLTrump" and "DrDenaGrayson" seem to be close, which again cannot be translated to semantic information. It is considered coincidental to be close but not coincidental to be in the same color-area (community).

In both plots (node diagrams Figs 17 and 18) it is not possible to keep readable names of each particular node, through the high density of nodes. The nodes in Fig 18 are the users on Twitter who have posted the tweets included in this plot and similarly, the nodes in Fig 17 are the users of YouTube which are channels. In order to highlight only the important nodes, we manage to provide the names for the most significant nodes based on the PageRank score.

For illustration purposes, nodes in both plots (17,18) with high degrees are shaped like circles (lightly highlighted circle-like scheme) in this plot, but are variables in Gephi that unfortunately cannot be explained semantically in this context and should be ignored.

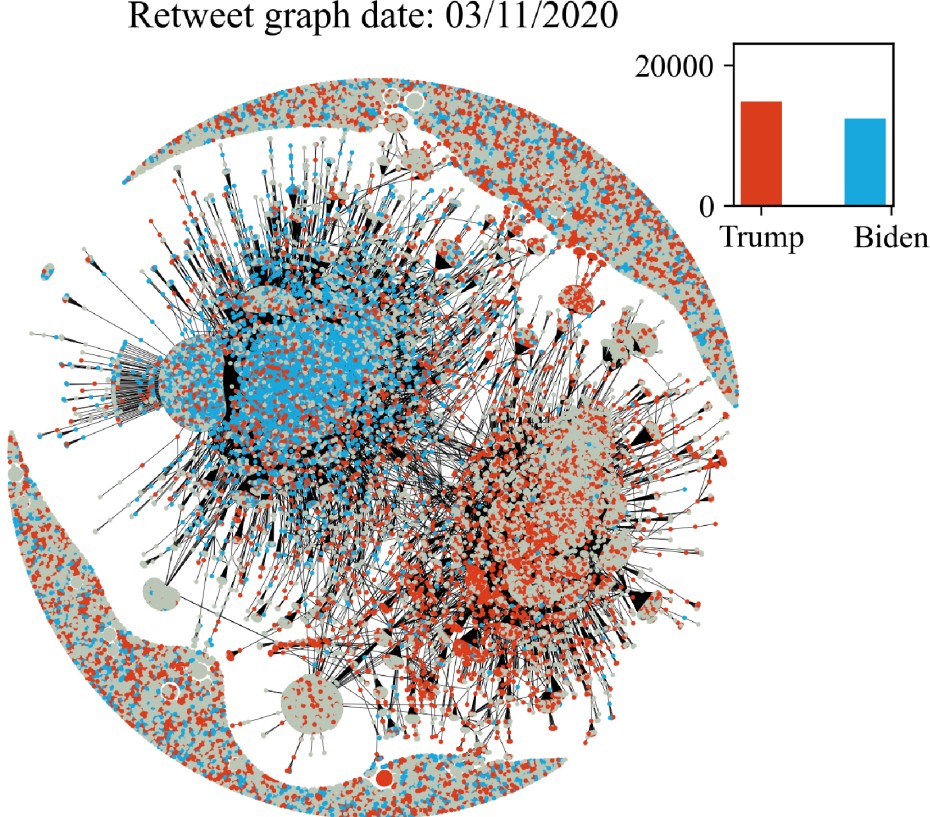

**Fig 11. Retweet graph relation between users in our dataset.** Colours represent entity activity by users. Red colour entity Trump and blue color entity Biden.

Fig 19 shows the interactions between the 3 largest YouTube communities (top half) in the YouTube-comment graph (YT) and the 6 largest Twitter communities in the Retweet graph (RT). Each community is named after its highest PageRank member (or the second highest, when more clear) in the corresponding graph. The size of each relation depicts the number of

**Table 3. Top retweeted users (highest indegree).**

| User | Number of Followings | Number of Followers |
|---|---|---|
| User 1 | 1.03K | 826.4K |
| User 2 | 5.1K | 2.7M |
| User 3 | 97 | 248.3K |

**Table 4. Top retweeted users (highest out degree).**

| User | Number of Followings | Number of Followers |
|---|---|---|
| User 1 | 4.7K | 5K |
| User 2 | 3.6K | 3.6K |
| User 3 | 5K | 8.1K |

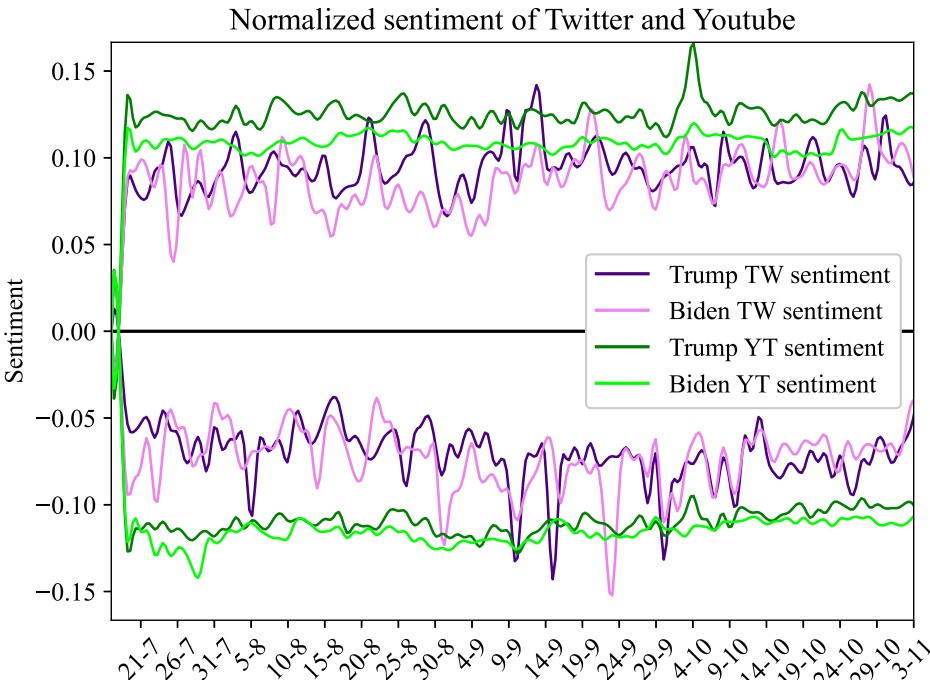

**Fig 12. Result of daily average sentiment per entity for Twitter and YouTube collected dataset.**

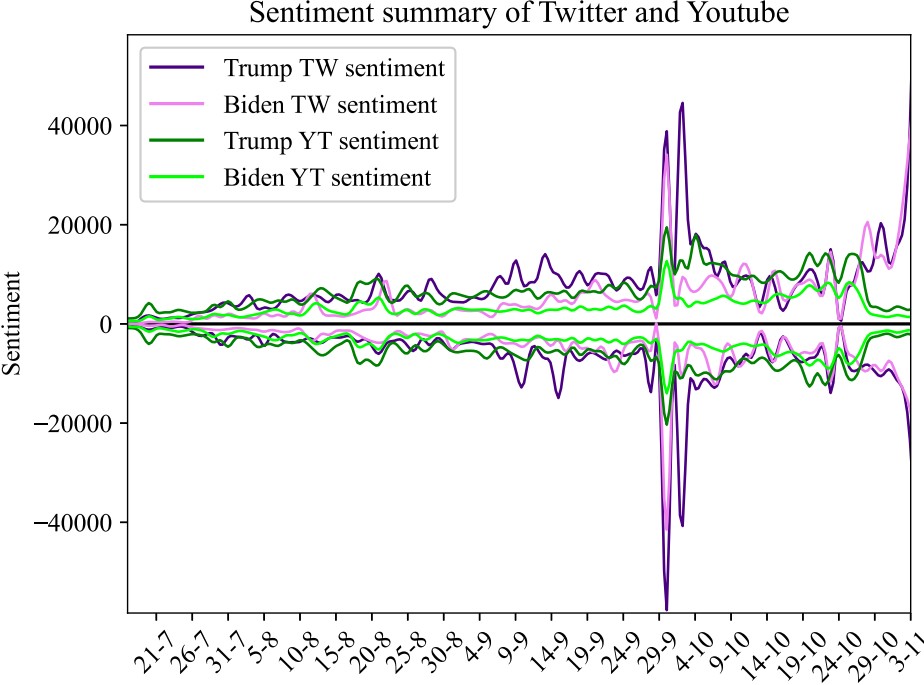

**Fig 13. Result of overall sentiment analysis for Twitter and YouTube.** Particular sentiment values present the overall sentiment that was posted by users at given dates.

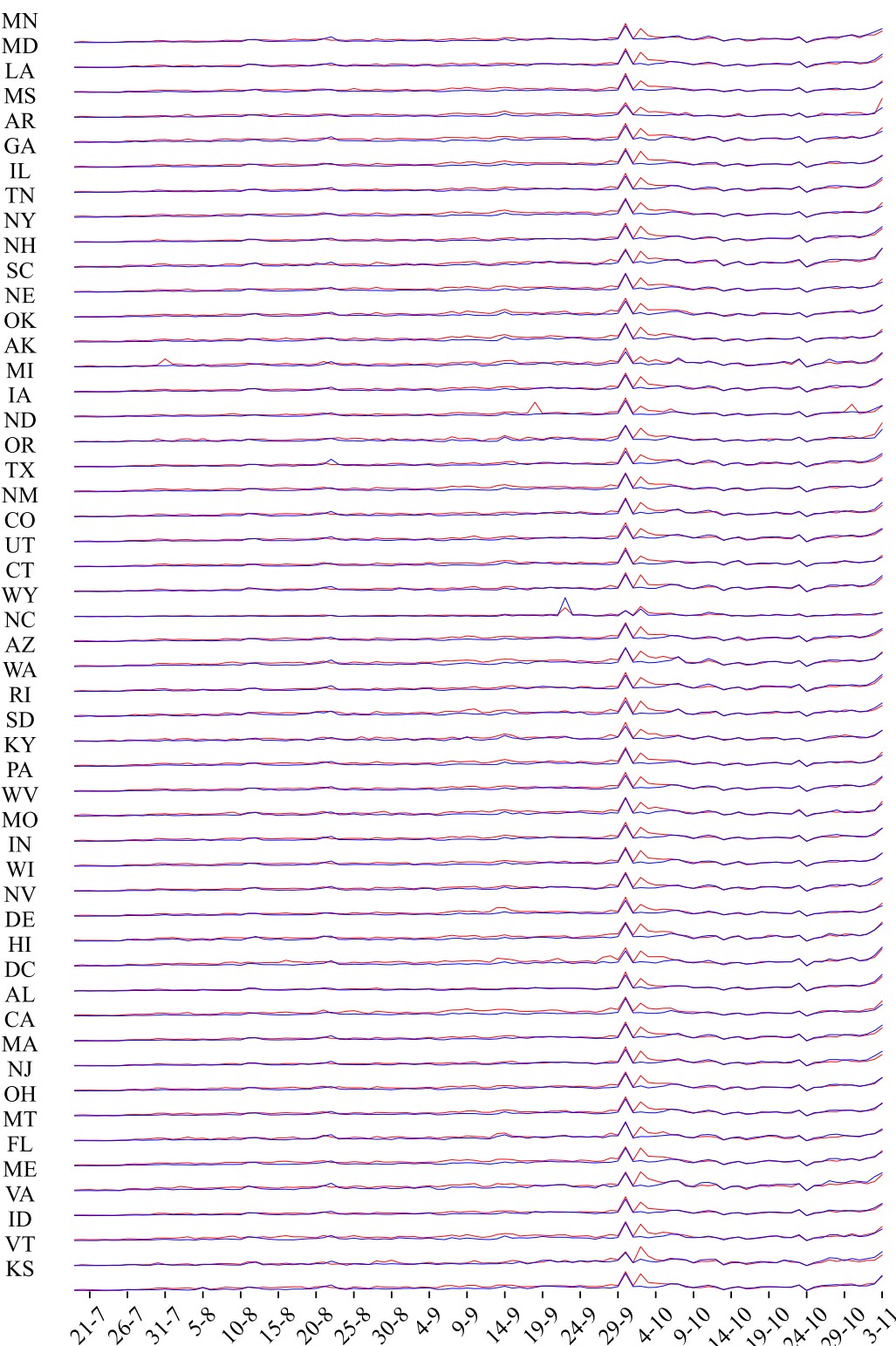

**Fig 14. Positive sentiment time series for the two sets of hashtags for each state.**

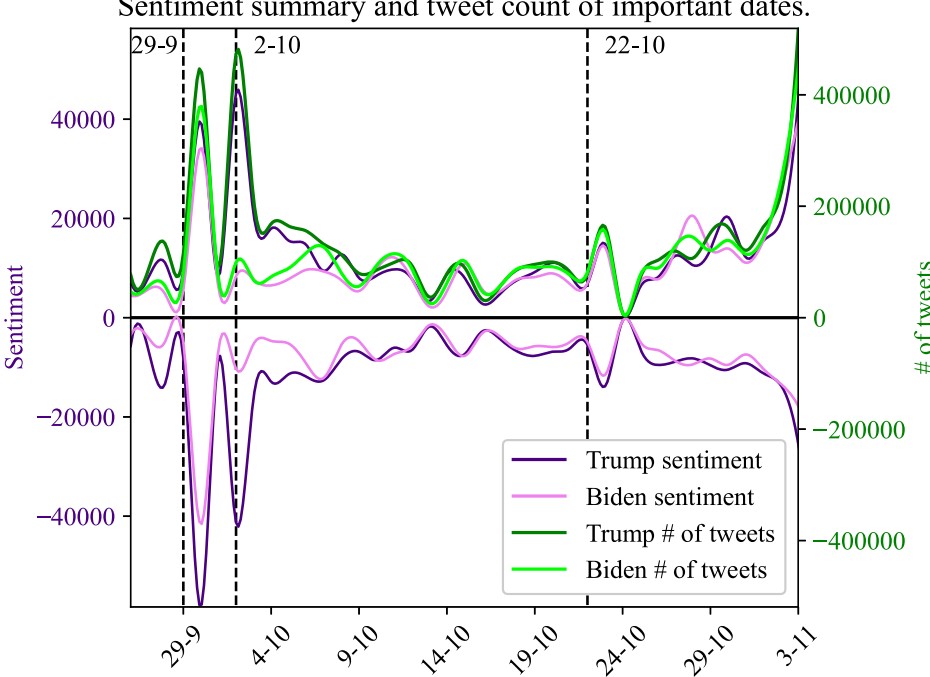

**Fig 15. Summary of user tweets sentiment values per each entity followed by the number of tweets.** In this plot, we mention 3 important dates in the dataset, where 29/09 is the date of the first debate, 2/10 the date when Donald Trump was tested positive for COVID-19, and 22/10 the date of the last election debate.

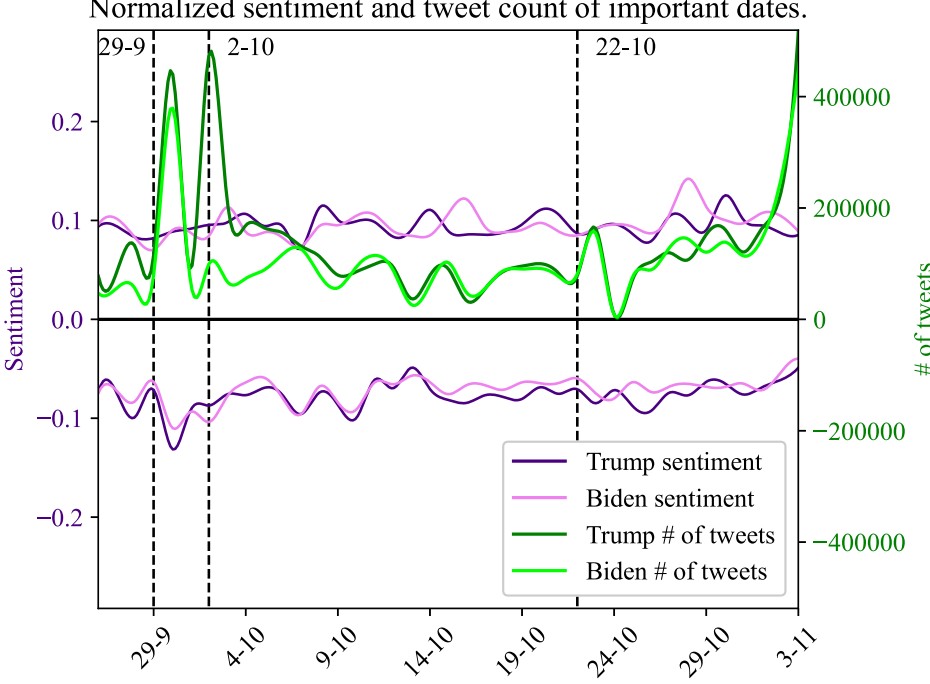

**Fig 16. Normalized sentiment values of user tweet per each entity followed by the number of tweets.** In this plot, we mention 3 important dates in the dataset, where 29/09 is the date of the first debate, 2/10 the date when Donald Trump was tested positive to COVID-19, and 22/10 the date of the last election debate.

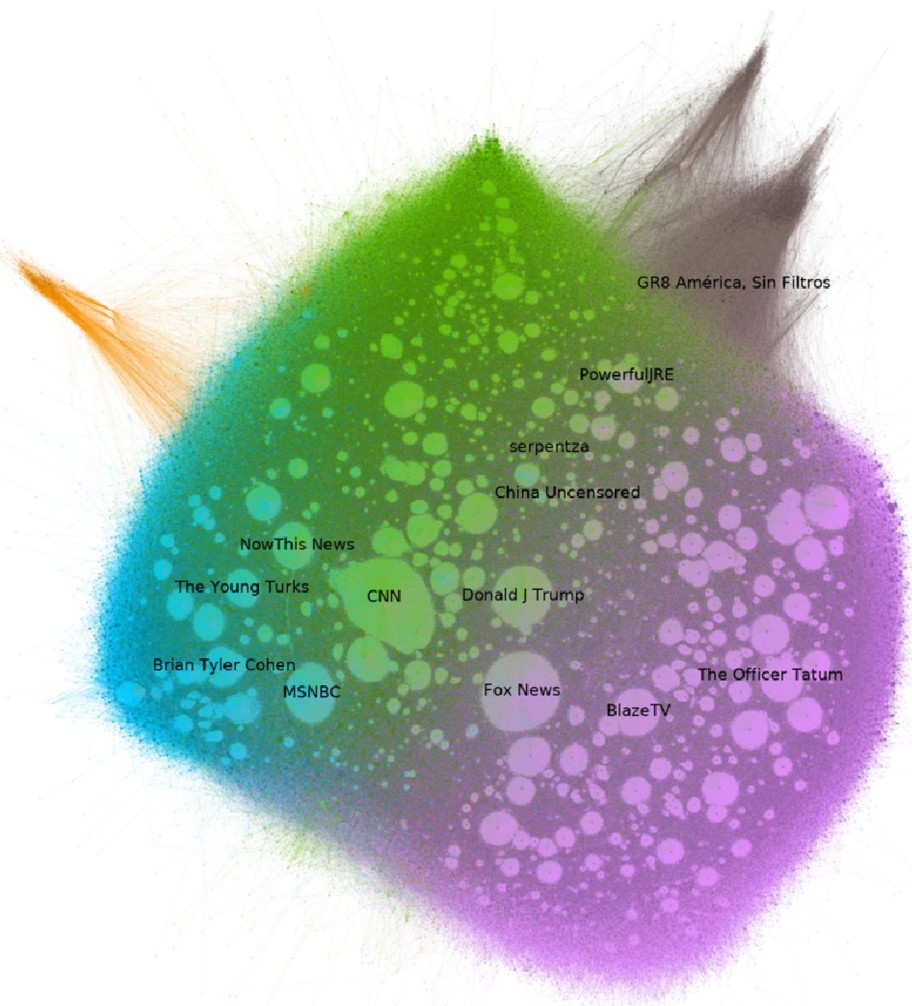

**Fig 17. The 3-core of the YouTube comment graph, color-coded by the community.**

users in the RT community, that tweeted video URLs from any channel in the YT community. As expected, communities affiliated with the Trump campaign on Twitter, mostly post video links from channels in the Fox News YouTube community, whereas members of the Twitter community mostly affiliated with the Biden campaign post mostly video links from YouTube channels in the CNN YouTube community.

One Twitter community that seems to follow an unexpected pattern is mostly formed by Arabic speakers who, in this setting, tend to use Twitter more as a news media and less as a social network for disseminating political opinions. This observation supports linking the Twitter Retweet graph with the YouTube comment graph using tweets of posted videos.

## 6 Conclusions

The purpose of this study is to shed light on the online discourse on social media, happening during the US presidential elections of November 2020. We focus on two main social media Twitter and YouTube. This work having obtained the tweets for the most popular hashtags

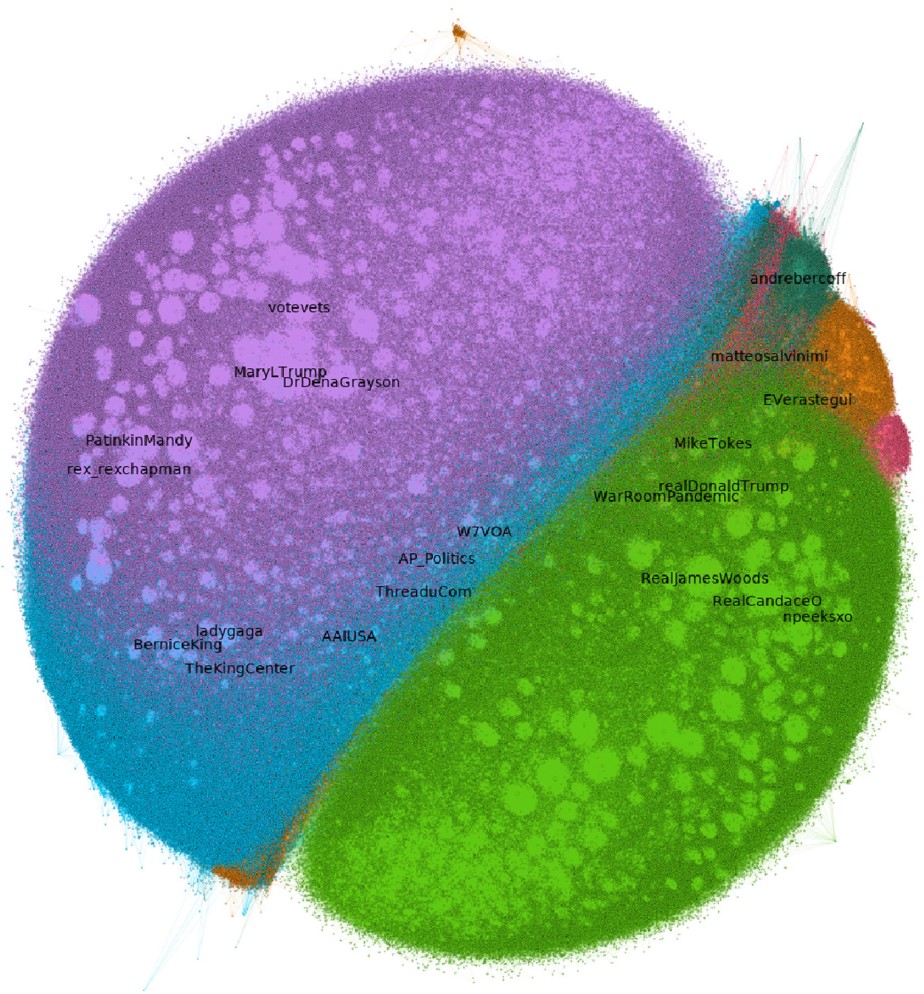

**Fig 18. The 3-core of the retweet graph, color-coded by community.**

regarding the US elections 2020, as well as the extracted unique YouTube videos, performs an analysis to study the connection between these two social networks, by following a series of steps including: volume analysis of tweets and users, identification of entities, and correlation between the features of the YouTube videos. Next, we apply sentiment analysis on the Twitter corpus and the YouTube metadata and show that the positive sentiment is higher for Donald Trump in comparison with Joe Biden. We identify how real-world events trigger user discussions on social media around the elections.

The next step is to study the Retweet graph across six different time points on the dataset, from July to September 2020, highlight the two main entities ('Biden' and 'Trump') and show how the main connected component becomes denser through time. Finally, the results from the previous steps of the analysis allow us to link the Twitter Retweet graph with the YouTube comment graph and show the interactions between the 3 largest YouTube communities, in the YouTube-comment graph and the 6 largest Twitter communities in the Retweet graph (RT). We include sarcasm detection in our future plans, since it will require a crowd-sourcing technique after the election period, in order to form an adequate ground truth dataset for the training process.

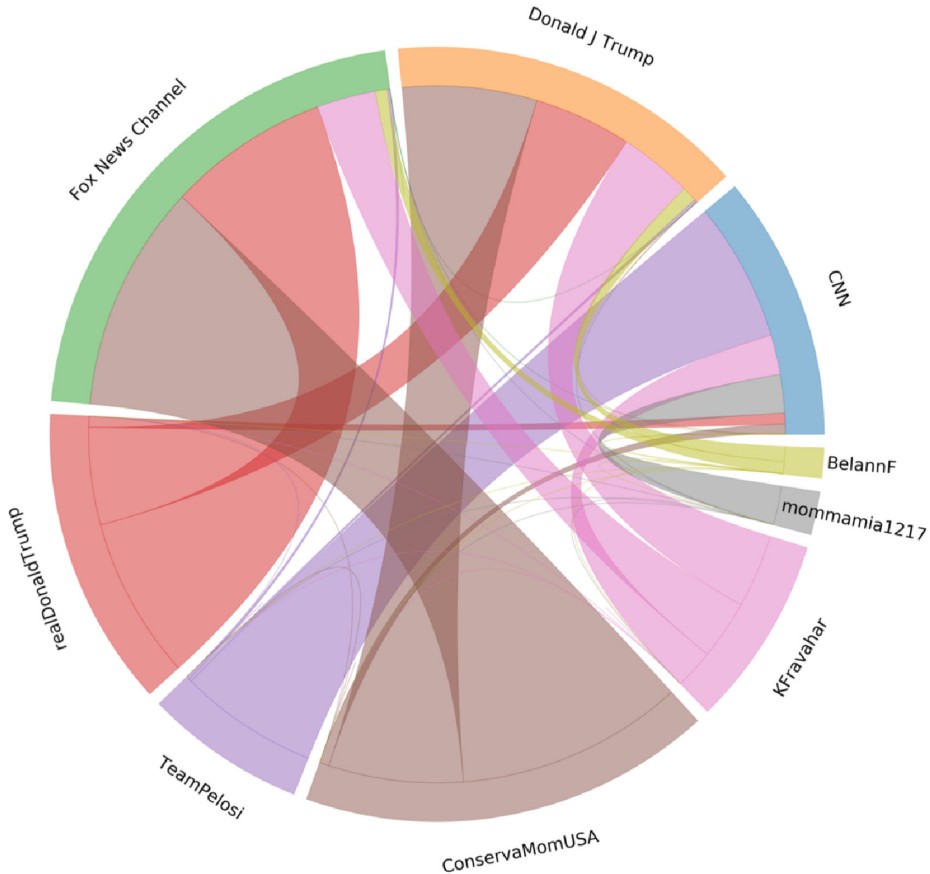

**Fig 19. Interactions between largest communities on YouTube and Twitter.**

## Supporting information

**S1 Appendix. List of all hashtags.**
(PDF)

**S1 Text.**
(TXT)

**S1 Data.**
(XLSX)

## Author Contributions

**Conceptualization:** Alexander Shevtsov, Maria Oikonomidou, Despoina Antonakaki, Polyvios Pratikakis.

**Data curation:** Alexander Shevtsov, Maria Oikonomidou, Despoina Antonakaki, Polyvios Pratikakis.

**Formal analysis:** Alexander Shevtsov, Maria Oikonomidou, Despoina Antonakaki, Polyvios Pratikakis.

**Funding acquisition:** Polyvios Pratikakis, Sotiris Ioannidis.

**Investigation:** Alexander Shevtsov, Maria Oikonomidou, Despoina Antonakaki, Polyvios Pratikakis.

**Methodology:** Alexander Shevtsov, Maria Oikonomidou, Despoina Antonakaki, Polyvios Pratikakis.

**Project administration:** Polyvios Pratikakis, Sotiris Ioannidis.

**Resources:** Polyvios Pratikakis, Sotiris Ioannidis.

**Software:** Alexander Shevtsov, Maria Oikonomidou, Polyvios Pratikakis.

**Supervision:** Despoina Antonakaki, Polyvios Pratikakis, Sotiris Ioannidis.

**Validation:** Alexander Shevtsov, Maria Oikonomidou, Despoina Antonakaki, Polyvios Pratikakis.

**Visualization:** Alexander Shevtsov, Maria Oikonomidou, Polyvios Pratikakis.

**Writing – review & editing:** Alexander Shevtsov, Maria Oikonomidou, Despoina Antonakaki, Polyvios Pratikakis.

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
