## [Decision Letter · Decision Letter 0]

3 Mar 2021

PONE-D-20-37937

What Tweets and YouTube comments have in common? Sentiment and Graph analysis on data related to US Elections 2020

PLOS ONE

Dear Dr. Antonakaki,

Thank you for submitting your manuscript to PLOS ONE. After careful consideration, we feel that it has merit but does not fully meet PLOS ONE’s publication criteria as it currently stands. Therefore, we invite you to submit a revised version of the manuscript that addresses the points raised during the review process.

We look forward to receiving your revised manuscript.

Kind regards,

Davide Bacciu

Academic Editor

PLOS ONE

Journal Requirements:

2. In your Methods section, please include additional information about your dataset and ensure that you have included a statement specifying whether the collection method complied with the terms and conditions for the website.

3.We note that you have indicated that data from this study are available upon request. PLOS only allows data to be available upon request if there are legal or ethical restrictions on sharing data publicly. For more information on unacceptable data access restrictions, please see http://journals.plos.org/plosone/s/data-availability#loc-unacceptable-data-access-restrictions.

4. Please ensure that you refer to Figure state pred wave.png in your text as, if accepted, production will need this reference to link the reader to the figure.

Additional Editor Comments:

The paper has been well received and the use of multiple social networks has been appreciated. Methodology appears adequate, though not innovative.

The manuscript requires a round of revision to better tune the claims in the paper and better substantiate the expected contributions with what is realistically demonstrated in the empirical part.

Reviewers' comments:

Reviewer's Responses to Questions

**Comments to the Author**

1. Is the manuscript technically sound, and do the data support the conclusions?

Reviewer #1: Partly

2. Has the statistical analysis been performed appropriately and rigorously? 

Reviewer #1: Yes

3. Have the authors made all data underlying the findings in their manuscript fully available?

Reviewer #1: Yes

4. Is the manuscript presented in an intelligible fashion and written in standard English?

Reviewer #1: No

5. Review Comments to the Author

Reviewer #1: -------

General summary

-------

The authors analyze the online discourse on the 2020 presidential elections in the United States. After collecting Twitter and YouTube data by supervised procedure, the authors present the results of various data-driven analyzes, including daily traffic, trends of hashtags, volume, sentiment and graph analysis. The work also focuses on studying the sentiment towards the two candidates and the effect of offline events on online social networks.

-------

Main successes and main problems

-------

The analyzes represent a twofold snapshot of a brief and well-defined historical moment at a fine-grained resolution. The work follows a logical and clear procedure, perhaps sometimes a little superficial.

The promises have been fulfilled and the results are what could be expected.

I appreciated the joint use of two different social networks, Twitter and YouTube, to observe differences of discussions and communities. Perhaps a brief consideration of the user bias of the two platforms could be added.

The methodological part reserved for sentiment analysis and community detection consists of the application of methods, i.e., Vader and Louvain, already implemented and documented. Consequently, the methodological contribution is limited.

-------

Major changes

-------

I would suggest to the authors to balance the presentation and contextualization of the arguments in favor of a correct valorisation of the work and distribution of expectations.

Section 3.5 appears not to be adequately supplemented, argued and defined. In addition to the Introduction, I would ask authors to contextualize the topic in Section 1.1. Check terminology.

The state of the art (Section 1.1.) mentions numerous recent articles. However, the literature is limited to sentiment analysis with hints of the consequences of real events on Twitter and YouTube data. I think it is necessary to contextualyze network analysis, e.g., theory/use of (characteristics of) networks and communities.

-------

Minor changes

-------

There are numerous food for thought that could enrich Section 3.2. These include emoticons, irony, swear words, bad language, youth language, and acronyms.

Please remove the claim of precedence from the abstract. In this case, the claim is unnecessary and hard to confirm.

-------

Small corrections

-------

- Capitalize “figure”, “table” and “section” when followed by a reference number, e.g., lines 106 and 155.

- Some missing spaces, e.g., line 71, 208, 203, captions of Tables 3 and 4.

6. PLOS authors have the option to publish the peer review history of their article (what does this mean?). If published, this will include your full peer review and any attached files.

Reviewer #1: No

---

## [Author Response · Author response to Decision Letter 0]

31 May 2021

The response to the review comments has been included in the rebutal letter, which has been uploaded as a separate file.

---

## [Decision Letter · Decision Letter 1]

30 Jul 2021

PONE-D-20-37937R1

What Tweets and YouTube comments have in common? Sentiment and Graph analysis on data related to US Elections 2020

PLOS ONE

Dear Dr. Antonakaki,

Thank you for submitting your manuscript to PLOS ONE. After careful consideration, we feel that it has merit but does not fully meet PLOS ONE’s publication criteria as it currently stands. Therefore, we invite you to submit a revised version of the manuscript that addresses the points raised during the review process.

The paper has been reviewed by an additional expert who is providing a quite detailed assessment of the key strenghts and weaknesses of the work. These should be carefully taken into consideration by the authors should they decide to send a revised manuscript, as the work will be checked against such open point. Namely, the review highlights that the paper tackles an interesting cross-platform approach and a potentially impactful application to political discourse. The main criticisms are instead directed towards an unsatisfactory consideration of the background work and the lack of a clear, homogenous, and strong articulation of the key points of innovation in the analysis proposed by this paper.

We look forward to receiving your revised manuscript.

Kind regards,

Davide Bacciu

Academic Editor

PLOS ONE

Journal Requirements:

Additional Editor Comments (if provided):

The paper has been reviewed by an additional expert who is providing a quite detailed assessment of the key strenghts and weaknesses of the work. These should be carefully taken into consideration by the authors should they decide to send a revised manuscript, as the work will be checked against such open point. Namely, the review highlights that the paper tackles an interesting cross-platform approach and a potentially impactful application to political discourse. The main criticisms are instead directed towards an unsatisfactory consideration of the background work and the lack of a clear, homogenous, and strong articulation of the key points of innovation in the analysis proposed by this paper.

Reviewers' comments:

Reviewer's Responses to Questions

**Comments to the Author**

1. If the authors have adequately addressed your comments raised in a previous round of review and you feel that this manuscript is now acceptable for publication, you may indicate that here to bypass the “Comments to the Author” section, enter your conflict of interest statement in the “Confidential to Editor” section, and submit your "Accept" recommendation.

Reviewer #2: (No Response)

2. Is the manuscript technically sound, and do the data support the conclusions?

Reviewer #2: Partly

3. Has the statistical analysis been performed appropriately and rigorously? 

Reviewer #2: No

4. Have the authors made all data underlying the findings in their manuscript fully available?

Reviewer #2: No

5. Is the manuscript presented in an intelligible fashion and written in standard English?

Reviewer #2: Yes

6. Review Comments to the Author

Reviewer #2: # Summary

This paper presents a primarily descriptive analysis on a pair of datasets collected from Twitter and YouTube during the 2020 US presidential election. Analyses include descriptions of the time series data extracted from both platforms, plots of several retweet-based interaction networks over time, sentiment analysis of messages and comments on Twitter and YouTube respectively, event identification, and Twitter- and YouTube-based community analysis. Results show temporal variation in volume and senitment and suggest some consistency across Twitter and YouTube.

# Strengths

This paper's major strength comes from its potential for cross-platform analysis. Extracting YouTube videos from Twitter content and comparing temporal and sentiment patterns across platforms can provide interesting insight about cross-platform and event-based influence.

# Weaknesses

The most significant weaknesses in this paper are its lack of clear contributions to the field and the (relatedly) limited connection it has with prior work studying political discourse in lead-up to elections on social media. For example, there's a huge body of literature on Twitter and politics in general [1], and the 2016 US presidential election specifically [2-3], so how does that work inform what we might expect to find in Twitter and YouTube during the 2020 election? Are any of the results identified in the four months leading up to Election Day in 2020 surprising given what we know about 2016? Even for 2020, other work has been done on YouTube and sentiment (e.g., Singh and Sikka [4]), so how consistent are these results with those from other papers in this area?

- While the paper's related work section does identify a number of related efforts in this area, more needs to be done to explain what *important* questions this paper answers that have not already been answered. That is, we need to know not just how this paper investigates a slightly different context than the other papers, but how those other papers inform this work. Does this work contradict existing results or expectations? Does the existing body of work contradict itself, and this paper resolves this contradiction? These are crucial questions that need to be answered to connect this work to a particular community and demonstrate what the most important result in this paper really is.

- Related to above, the paper lacks a clear set of hypotheses about what we might expect in this context. This work is not exploratory given the volume of work in this area, and providing a set of hypotheses based on the literature would help us understand the important results in this work.

Beyond this contextualization, data collection and validity of the data collected in this effort are unclear. How exactly was this dataset collected from Twitter? I understand the API was used, but was the streaming API used or the retrospective API? If the collection was built around particular keywords, what were those keywords, and how were they selected? The paper mentions a list of hashtags, but it appears this list is extracted from the collected dataset, not a set of tags used to develop the dataset. We need additional detail about how this dataset was collected to assess what is missing. For example, popular hashtags like #StopTheSteal, #HidingBiden, or other partisan hashtags do not appear in this data.

- To address this issue, the paper should clearly explain exactly how the data was collected and why this timeframe is reasonable.

The paper also claims to present an analysis of sentiment per US state, but no information is provided about *how* this analysis maps messages to US states. Geolocation in social media is a known and open problem, and inclusion of this analysis necessitates a description of how this geolocation is performed.

Other comments:

- The analysis of retweet graphs are primarily qualitative and lack a compelling frame. What did we expect to see here? How do we know the differences observed between the two candidates is meaningful?

- Sentiment analysis in Vader is not specifically built for social media; the paper should spend some effort demonstrating its assessment of sentiment towards the presidential candidates is valid.

- More detail is needed for entity extraction. Is the search for trump/biden a direct token match or a substring match? E.g., would it match a tweet with #TeachersForBiden? Also, for tweets with tags like #VoteBlue, they may be specific to Biden but not directly mention him; how does that potentiality impact analyses?

- In general, node-edge diagrams are not particularly useful visualizations. Additional effort is needed to describe what is interesting about the communities in these figures, what size means, why certain nodes are labeled and others note, etc.

- The paper is missing a "Data Availability Statement" in the text.

# Final Comments

I applaud the authors for engaging with an important set of issues (namely political communications in online spaces), and I especially like the potential for cross-platform analyses and connections.

That said, the paper presents an exploratory and descriptive piece about a topic that has a lot of background, and the paper does not sufficiently engage with this background. Given the volume of work in this area, some of which overlaps with this work, the current version of the paper does not have a clear path toward a major contribution. Instead, it feels like the paper is primarily a collection of different, unconnected analyses that lack clear motivation.

I encourage the authors to continue this work and find ways to build on the prior background in this space.

# Related Work

[1] BARBERÁ, P., CASAS, A., NAGLER, J., EGAN, P., BONNEAU, R., JOST, J., & TUCKER, J. (2019). Who Leads? Who Follows? Measuring Issue Attention and Agenda Setting by Legislators and the Mass Public Using Social Media Data. American Political Science Review, 113(4), 883-901. doi:10.1017/S0003055419000352

[2] Sahly A, Shao C, Kwon KH. Social Media for Political Campaigns: An Examination of Trump’s and Clinton’s Frame Building and Its Effect on Audience Engagement. Social Media + Society. April 2019. doi:10.1177/2056305119855141

[3] Zhang Y, Wells C, Wang S, Rohe K. Attention and amplification in the hybrid media system: The composition and activity of Donald Trump’s Twitter following during the 2016 presidential election. New Media & Society. 2018;20(9):3161-3182. doi:10.1177/1461444817744390

[4] S. Singh and G. Sikka, "YouTube Sentiment Analysis on US Elections 2020," 2021 2nd International Conference on Secure Cyber Computing and Communications (ICSCCC), 2021, pp. 250-254, doi: 10.1109/ICSCCC51823.2021.9478128.

7. PLOS authors have the option to publish the peer review history of their article (what does this mean?). If published, this will include your full peer review and any attached files.

Reviewer #2: No

---

## [Author Response · Author response to Decision Letter 1]

12 Oct 2021

Response to the reviewers

Review comment 1) The most significant weaknesses in this paper are its lack of clear contributions to the field and the (relatedly) limited connection it has with prior work studying political discourse in lead-up to elections on social media. For example, there's a huge body of literature on Twitter and politics in general [1], and the 2016 US presidential election specifically [2-3], so how does that work inform what we might expect to find in Twitter and YouTube during the 2020 election? Are any of the results identified in the four months leading up to Election Day in 2020 surprising given what we know about 2016? Even for 2020, other work has been done on YouTube and sentiment (e.g., Singh and Sikka [4]), so how consistent are these results with those from other papers in this area?

Abstract, Introduction and Background work have been reformed and as we have included a lot of cross-platform studies. 

We have also included the papers mentioned in the review comment, among other new background work that we have added. [1] is referenced in subsection: “Political content analysis on Twitter”, of section “Background”. [2] is referenced in the first paragraph of “background”, [3] is referenced in the first paragraph of subsection “Political content analysis on Twitter” in “Background” and [4] is referenced in the first paragraph of subsection “Content analysis on YouTube” in “Background”. 

Regarding the results in the four months leading to election day in 2020 from our analysis, cannot correlate with previous work done for 2016. The nature of the two analyses are different and the dataset in the previous studies (in 2016) seem to be limited , in terms of number of tweets as well as covering multiple social media. In these terms we cannot perform a comparison between the two studies. More specifically in [1 BARBERÁ, P., CASAS et al. ] they use a dataset of tweets sent by the members of the 113th House and Senate of the US Congress (2013–14), while our dataset contains tweets from the popular hashtags available, which means they could be sent by users (including electorate) and potentially members of the House and Senate. Additionally, they do not include data from other social networks (like YouTube in our study) and the pipeline of the analysis is completely different (e.g. they apply probabilistic model LDA, and we are not applying topic analysis at all). 

The same applies for the related studies in 2020 (e.g., Singh and Sikka [4]). Specifically, they are applying an analysis on 200 YouTube comments, while our analysis includes 20M tweets and the comments of 29K YouTube videos. The nature of the two analyses is different in many perspectives, since we apply a comparative analysis between the two social networks (Twitter and Youtube) and sentiment analysis is just a step of the pipeline necessary towards this goal, while authors in [4] are solely focusing on the sentiment analysis of the YouTube comments. 

Finally, regarding [2] what they study is more close to our analysis, except that they are comparing Twitter and Facebook, while our analysis focuses on Twitter and YouTube. Also, they focus the emotional frames while we demonstrate how the communities correlate between Twitter and YouTube. 

[The last two paragraphs are not included in the manuscript, since it is a response to the review and out of the context of the study]. 

-Review comment 2) While the paper's related work section does identify a number of related efforts in this area, more needs to be done to explain what *important* questions this paper answers that have not already been answered. That is, we need to know not just how this paper investigates a slightly different context than the other papers, but how those other papers inform this work. Does this work contradict existing results or expectations? Does the existing body of work contradict itself, and this paper resolves this contradiction? These are crucial questions that need to be answered to connect this work to a particular community and demonstrate what the most important result in this paper really is.

Related to above, the paper lacks a clear set of hypotheses about what we might expect in this context. This work is not exploratory given the volume of work in this area, and providing a set of hypotheses based on the literature would help us understand the important results in this work.

We have reformed the paper including the abstract, the introduction, and the background work, highlighting our reshaped contributions that focus on the correlation and interactions between Twitter and YouTube communities, which actually is our novelty.

-Review comment 3) Beyond this contextualization, data collection and validity of the data collected in this effort are unclear. How exactly was this dataset collected from Twitter? I understand the API was used, but was the streaming API used or the retrospective API? If the collection was built around particular keywords, what were those keywords, and how were they selected? The paper mentions a list of hashtags, but it appears this list is extracted from the collected dataset, not a set of tags used to develop the dataset. We need additional detail about how this dataset was collected to assess what is missing. For example, popular hashtags like #StopTheSteal, #HidingBiden, or other partisan hashtags do not appear in this data.

To address this issue, the paper should clearly explain exactly how the data was collected and why this timeframe is reasonable.

We address this comment in the “Dataset” section, where Twitter and YouTube dataset collection procedure is described.

Considering the hashtags used for our analysis, we only show in the manuscript the list of 20 most popular hashtags in our dataset. The entire list contains 585.486 unique entries of hashtags, which is too long to be added to the manuscript and is included in the submission as a separate file (all_hashtags.txt) which contains all the HTs used and the number of tweets corresponding in each HT that we have in our dataset.

We added in the manuscript the following paragraph:

We considered that this date was a reasonable starting point for collecting our dataset since the amount of tweets in the corresponding hashtags we collect begin to accumulate a significant number, as well as the semantic of the content, started to be more relevant to the conversation related to the elections.

Regarding the completeness of the content covered by our dataset, we acknowledge the fact that additional minor hashtags may exist during that period that were not crawled and included in our corpus. We consider our dataset a complete online discourse since we got the majority of the hashtags available in that period before the elections. Additionally, there was an overlap between the hashtags on the election discourse and these hashtags were cross referenced in the tweets. For example, some tweets included the popular hashtags (\\#Vote) and the minor hashtags were also mentioned in the same tweet. This tweet is included in our dataset because of \\#Vote. Additionally, we included only the hashtags that we general and not in favor of a particular candidate. We try to include only two hashtags that are in favor of each candidate, keep a balance between them and not introduce a bias towards one of them. Finally, the main amount of the political conversation was gathered in the popular hashtags and the rest of them do not contain a significant amount of tweets. In the appendix, in figure \\ref{table:allhashtags} we show the list of 20 most popular hashtags in our dataset. The entire list contains 585.486 unique entries of hashtags, which is too long to be added to the manuscript. 

We also included in our submission the whole list of HTs crawled for this dataset. [filename : all_hashtags.txt]

-Review comment 4) The paper also claims to present an analysis of sentiment per US state, but no information is provided about *how* this analysis maps messages to US states. Geolocation in social media is a known and open problem, and the inclusion of this analysis necessitates a description of how this geolocation is performed.

This comment is addressed in the “Sentiment Analysis” section.

As it appears in the manuscript:

In order to identify the location of the user accounts, we extract this information from the corresponding field named 'location' in the Twitter user object. Additionally, we append this information with the location field in the tweet object of the Twitter API. We acknowledge the fact that the first field may be not updated by the user or the second field could be missing because the user did not approve to share the location. This means that the location for many users could be missing. Nevertheless, this is the closest information we have from Twitter about the user location.} We notice the daily fluctuations for every state per entity (blue is the entity 'Biden' and red is for 'Trump'). The juxtaposition of the time series in the form resembling an EEG makes it easier to discern localized events from nation-wide Twitter traffic.

The list of state abbreviations can be found here: \\cite{states_abbr}.

-Review comment 5) The analysis of retweet graphs are primarily qualitative and lack a compelling frame. What did we expect to see here? How do we know the differences observed between the two candidates is meaningful?

As written also in the manuscript:

The retweet graphs show the volume of the tweets and the relation between the nodes which represent the users. This apposition of the graphs through the whole pre-elections period, demonstrates how the volume of the communication and interaction between the involved users evolves. The results were expected as we see the density of graph increasing, which means the volume of the tweets increases, and the discourse is getting more intense, as we approach the period close to the elections.

The discussion around each candidate evolves through time as well. The corresponding hashtag for each candidate is mentioned as long as the electorate references them in the discourse. Of course, this does not necessarily means a preference towards Trump or Biden. As mentioned, as well in section \\ref{dataset}, this analysis does not focus on the prediction of the preference of the electorate or the outcome of the elections, we expect to see how the users engage in the online conversation. As we get closer to the election date, we notice some components being formed in the graph that show the conversations about each candidate and the conversation about both of them in figure 11 [or in \\ref{retweet_graph}] in the last bottom sub-graph (f).

Additionally, we see two main graph components throughout the whole period that change colour according to the daily events. For example, in the button left subgraph (e) we notice that the colour is red in both components while the button right figure (f) one is painted blue (from J. Biden) . This happens because the left graph highlights the fact that Trump was infected with COVID-19 on 2/10, and the conversation about Trump was more intense and increased in terms of tweets. Also in the middle right plot we show the day after the debate, which mixed red-blue colour makes sense if we consider the conversations this particular debate initiated. The colours as expected seem to resume back to normal on the last bottom right (3/11, which correspond to the main two components blue and red, one for each candidate, since on this date none of the candidates seem to attract any peculiar attention.

-Review comment 6) Sentiment analysis in Vader is not specifically built for social media; the paper should spend some effort demonstrating its assessment of sentiment towards the presidential candidates is valid.

As justified in the text as well :

There is a plethora of previous studies as mentioned in section \\ref{Background} that analyze the sentiment in Twitter. Vader is broadly used in this domain as shown in \\cite{zahoor2020TWSent, elbagir2019twitter, pano2020Complete, ramteke2016election, shelar2018sentiment, park2018sentiment, mustaqim2020twitter, bose2021survey, al2020suspicious, yaqub2020tweeting, alharbi2019twitter}.

-Review comment 7) More detail is needed for entity extraction. Is the search for trump/biden a direct token match or a substring match? E.g., would it match a tweet with #TeachersForBiden? Also, for tweets with tags like #VoteBlue, they may be specific to Biden but not directly mention him; how does that potentiality impact analyses?

As described in section “The Entities and their sentiment” of the manuscript, the entities of J. Biden and D. Trump were identified with a computed set of keywords (including the hashtags and enriched with the candidates’ names and the parties they are representing). The selected keywords are shown in table ref{table:allentities} [Table 2: The complete list of all entities with the corresponding keywords that were used for each one.] . For example hashtag \\#VoteBlue, as a text does not contain the word Biden, but is associated with J. Biden campaign, and we manage to recognize it because of our keywords shown in table 2. 

For this reason, we use as keywords the hashtags correlated with each candidate. This approach increases the accuracy of the association of a particular tweet/user with the described entities.

\\The keywords are being searched in lower case in order to avoid any misspelling and user upper-lower case writing style.

-Review comment 8) In general, node-edge diagrams are not particularly useful visualizations. Additional effort is needed to describe what is interesting about the communities in these figures, what size means, why certain nodes are labeled and others note, etc.

This is explained in detail in the manuscript in section 5 Linking Twitter and YouTube data:

Fig17.eps illustrates the 3-core YouTube comment graph that represents the relation of which channel has commented on which channel. Different colored areas (purple, light grey, green blue) in the graph represent different communities that are formed between the YouTube channels produced by the Louvain algorithm. For example purple indicates the community between the “Blaze TV”, “The Officer Tatum” and the “ Fox News”. However “ Fox News” is also between the purple and the grey which shows the next community (formed by “Fox News” and “Donald J Trump”). These communities do not necessarily correspond to a real life community are a result of the Louvain algorithm of Gephi. 

Regarding the labels, in this figure we also show the 13 most popular channels after running PageRank.

Similarly, in plot 18 we show the 3-core Retweet graph that represents the relation which users have retweeted. In the graph we notice seven colored areas of different size, indicating different communities, that as in plot 17, do not necessarily correspond to a real life community. Also the text labels shown, for example “MaryLTrump” and “DrDenaGrayson” seem to be close, which again cannot be translated to semantic information. It is considered coincidental to be close but not coincidental to be in the same color-area (community). 

In both plots (node diagrams 17.eps-18.eps) it is not possible to keep readable names of each particular node, through the high density of nodes. The nodes in Fig18.eps are the users in Twitter who have posted the tweets included in this plot and similarly the nodes in Fig17.eps are the users of YouTube which are channels. In order to highlight only the important nodes we manage to provide the names for most significant nodes based on the PageRank score.

For illustration purposes nodes in both plots (17,18) with high degree are shaped like in circles (lightly highlighted circle-like scheme) in this plot, but are variables in Gephi that unfortunately, cannot be explained semantically in this context and should be ignored.

-Review comment 9) The paper is missing a "Data Availability Statement" in the text.

The dataset is available through zenodo and this was already described in subsection: “How to obtain the dataset”, which has been moved to subsection: “Data Availability” subsection located in the dataset section.

https://zenodo.org/record/4618233#.YGGJU2Qzada

---

## [Decision Letter · Decision Letter 2]

7 Mar 2022

PONE-D-20-37937R2

What Tweets and YouTube comments have in common? Sentiment and Graph analysis on data related to US Elections 2020

PLOS ONE

Dear Dr. Antonakaki,

Thank you for submitting your manuscript to PLOS ONE. After careful consideration, we have decided that your manuscript does not meet our criteria for publication and must therefore be rejected.

Specifically:

I am sorry that we cannot be more positive on this occasion, but hope that you appreciate the reasons for this decision.

Yours sincerely,

Ayan Seal, Ph.D

Academic Editor

PLOS ONE

Reviewers' comments:

Reviewer's Responses to Questions

**Comments to the Author**

1. If the authors have adequately addressed your comments raised in a previous round of review and you feel that this manuscript is now acceptable for publication, you may indicate that here to bypass the “Comments to the Author” section, enter your conflict of interest statement in the “Confidential to Editor” section, and submit your "Accept" recommendation.

Reviewer #3: (No Response)

2. Is the manuscript technically sound, and do the data support the conclusions?

Reviewer #3: No

3. Has the statistical analysis been performed appropriately and rigorously? 

Reviewer #3: No

4. Have the authors made all data underlying the findings in their manuscript fully available?

Reviewer #3: Yes

5. Is the manuscript presented in an intelligible fashion and written in standard English?

Reviewer #3: No

6. Review Comments to the Author

Reviewer #3: The author needs to identify the novelty in the paper, as sentiment analysis is a vast field-. A lot of work has already been done in this field. The work is not up-to the mark lagging novelty .

Also their are statistical measures to show that your work is better than other works which is basically lagging in authors work

No comparative study is done to depict that your work ix better than other work.

Above all the paper seems to lag a lot of novelty and hence It is not accepted

7. PLOS authors have the option to publish the peer review history of their article (what does this mean?). If published, this will include your full peer review and any attached files.

Reviewer #3: No

- - - - -

---

## [Author Response · Author response to Decision Letter 2]

13 May 2022

This submission is a response to the "PLOS ONE: Your Appeal for the manuscript (PONE-D-20-37937R2) - [EMID:87b2b482bf78e2ed]"

After the final round of reviews, the manuscript has been handled to a new Editor, and as address in the response to reviewers: file PLOS Fourth review (rejection).pdf we responded with the request to be revisited. 

In this submission we include the cover letter, all the necessary latex files (plos_latex_third_review.tex) and images in order to produce the manuscript, as well as: 

The manuscript in pdf with highlighted changes from the first submission (Revised Manuscript with Track Changes_25_august_2021.pdf) the second submission (Revised Manuscript with Track Changes_12_October_.pdf)

---

## [Editor Report · Decision Letter 3]

14 Jun 2022

What Tweets and YouTube comments have in common? Sentiment and Graph analysis on data related to US Elections 2020

PONE-D-20-37937R3

Dear Dr. Antonakaki,

We are pleased to inform you that your manuscript has been judged scientifically suitable for publication and will be formally accepted for publication once it meets all outstanding technical requirements.

**I acknowledge that the authors face several major challenges in terms of the number of revisions and disparity of review comments across the three versions of this manuscript. I commend the authors for their resilience and diligence in fully addressing all relevant comments with suitable content while also maintaining the main focus of this paper. Specifically, the challenges of homophily, echo chambers are increasingly prominent on social media platforms, and this analysis of such behaviors within and across platforms during a "social media significant"  event such as the US Elections of 2020" is practical and meaningful contribution to research literature. I also acknowledge  this manuscript aligns with the Plos One publication principles in conveying this contribution to the wider community. **

Kind regards,

Daswin De Silva

Academic Editor

PLOS ONE

Additional Editor Comments (optional):

Reviewers' comments: 

No further reviewer comments. I commend the authors in addressing all reviewer comments effectively, across the past three review cycles. The manuscript has already been significantly revised and now much more relatable to the Plos One readership.

---

## [Editor Report · Acceptance letter]

27 Jun 2022

PONE-D-20-37937R3 

What Tweets and YouTube comments have in common? Sentiment and Graph analysis on data related to US Elections 2020 

Dear Dr. Antonakaki:

I'm pleased to inform you that your manuscript has been deemed suitable for publication in PLOS ONE. Congratulations! Your manuscript is now with our production department. 

Kind regards, 

on behalf of

Dr. Daswin De Silva 

Academic Editor

PLOS ONE